# Decoding the activated stem cell phenotype of the neonatally maturing pituitary

**Emma Laporte[1], Florian Hermans[1,2], Silke De Vriendt[1], Annelies Vennekens[1], Diether Lambrechts[3,4], Charlotte Nys[1], Benoit Cox[1], Hugo Vankelecom[1]***

[1]Laboratory of Tissue Plasticity in Health and Disease, Cluster of Stem Cell and Developmental Biology, Department of Development and Regeneration, KU Leuven, Leuven, Belgium; [2]Laboratory of Morphology, Biomedical Research Institute, Hasselt University, Diepenbeek, Belgium; [3]Center for Cancer Biology, VIB, Leuven, Belgium; [4]Laboratory for Translational Genetics, Department of Human Genetics, KU Leuven, Leuven, Belgium

**\*For correspondence:**
hugo.vankelecom@kuleuven.be

**Competing interest:** The authors declare that no competing interests exist.

**Abstract** The pituitary represents the endocrine master regulator. In mouse, the gland undergoes active maturation immediately after birth. Here, we in detail portrayed the stem cell compartment of neonatal pituitary. Single-cell RNA-sequencing pictured an active gland, revealing proliferative stem as well as hormonal (progenitor) cell populations. The stem cell pool displayed a hybrid epithelial/mesenchymal phenotype, characteristic of development-involved tissue stem cells. Organoid culturing recapitulated the stem cells' phenotype, interestingly also reproducing their paracrine activity. The pituitary stem cell-activating interleukin-6 advanced organoid growth, although the neonatal stem cell compartment was not visibly affected in $Il6^{-/-}$ mice, likely due to cytokine family redundancy. Further transcriptomic analysis exposed a pronounced WNT pathway in the neonatal gland, shown to be involved in stem cell activation and to overlap with the (fetal) human pituitary transcriptome. Following local damage, the neonatal gland efficiently regenerates, despite absence of additional stem cell proliferation, or upregulated IL-6 or WNT expression, all in line with the already high stem cell activation status, thereby exposing striking differences with adult pituitary. Together, our study decodes the stem cell compartment of neonatal pituitary, exposing an activated state in the maturing gland. Understanding stem cell activation is key to potential pituitary regenerative prospects.

## Editor's evaluation

This is a well conducted study on development of neonatal mouse pituitary using multiple ScRNA Seq and organoid culture models. In this revised version, the authors have done a thorough job of addressing the concerns of the reviewers on the previous version and significantly improved it. In particular they have provided additional data with regard to the role of Wnt signaling and further defined the IL6 null mouse phenotypes.

## Introduction

As central hub of the endocrine system, the pituitary plays a quintessential role in governing the endocrine glands throughout the body, thereby regulating key physiological processes including growth, metabolism, fertility, stress, and immunity. To execute this primordial function, the pituitary has to properly develop into a compound gland of multiple endocrine cells, encompassing somatotropes

**eLife digest** The pituitary gland is a pea-sized structure found just below the brain that produces hormones controlling everything from growth and stress to reproduction and immunity. To perform its role, the pituitary gland needs specialised hormone-producing cells, but it also contains stem cells. These stem cells can divide to produce more cells like themselves, or differentiate into cells of different types, including hormone-producing cells.

In mice, the stem cells of the pituitary gland appear to be activated in the first few weeks after birth, and later become 'quiescent' (or lazy) in the adult pituitary gland. However, it remains unclear how the activated state found in the maturing gland is established and regulated.

To answer this question, Laporte et al. used single-cell RNA sequencing, a technique that allows researchers to profile which genes are active in individual cells, which can provide vital information about the state and activity of a tissue. The researchers compared the cells of the maturing pituitary gland of newborn mice to the cells in the established gland of adult mice. This analysis revealed that the maturing pituitary gland is a dynamic tissue, with populations of cells that are actively dividing (including the stem cells), which the mature pituitary gland lacks. Additionally, Laporte et al. established the molecular basis for the activated state of the stem cells in the maturing pituitary gland, which relies on the activation of a cell signalling pathway called WNT.

To confirm these findings, Laporte et al. used an organoid system that allowed them to recapitulate the stem cell compartment of the maturing pituitary gland in a dish. When Laporte et al. blocked WNT signalling in these organoids, the organoids failed to form or divide. Furthermore, blocking the pathway directly in newborn mice reduced the number of dividing stem cells in the pituitary gland. Both findings support the notion that WNT signalling is required to establish the activated state of the maturing pituitary gland in newborn mice.

Laporte et al. also wanted to know whether the newborn pituitary gland responded to injury differently than the adult gland. It had already been established that the adult pituitary stem cells become activated upon injury, and that the gland has some regenerative capacity. However, when Laporte et al. injured the newborn pituitary gland, the gland was able to fully regenerate, despite the stem cells not becoming more activated. This is likely because these cells are already activated (or 'primed'), and do not require further activation to divide and repair the gland with the help of other proliferating cells.

With these results, Laporte et al. shed light on the activated state of the stem cells in the pituitary gland of newborn mice. This provides insight into the role of these stem cells, as well as unveiling possible routes towards regenerating pituitary tissue. This could eventually prove useful in medicine, in cases when the pituitary gland is damaged or removed.

(producing growth hormone [GH]), corticotropes (adrenocorticotropic hormone [ACTH]), lactotropes (prolactin [PRL]), gonadotropes (luteinizing hormone [LH] and/or follicle-stimulating hormone [FSH]), and thyrotropes (thyroid-stimulating hormone [TSH]) (*Melmed, 2011*; *Willems and Vankelecom, 2014*). In the mouse, the pituitary undergoes an intense growth and maturation process during the first postnatal weeks, with number and size of hormone-producing cells substantively expanding following proliferation of committed (progenitor) and endocrine cells (*Carbajo-Pérez and Watanabe, 1990*; *Laporte et al., 2021*; *Sasaki, 1988*; *Taniguchi et al., 2002*; *Zhu et al., 2015*). Simultaneously, the local stem cell (SOX2$^+$) compartment shows signs of activation including elevated abundance and expression of stemness pathways when compared to the adult pituitary stem cells (*Gremeaux et al., 2012*; *Laporte et al., 2021*). The stem cells can give rise to new endocrine cells, a property which has been found most prominent (although not extensive) during this neonatal period (*Andoniadou et al., 2013*; *Rizzoti et al., 2013*; *Zhu et al., 2015*). In contrast, stem cells in the mature, adult pituitary are quiescent and do not highly contribute to new endocrine cells during the (slow) homeostatic turnover of the gland (*Andoniadou et al., 2013*; *Laporte et al., 2021*; *Rizzoti et al., 2013*; *Vankelecom and Chen, 2014*). However, the adult stem cells become activated following local injury in the gland, showing a proliferative reaction and signs of differentiation toward the ablated cells, coinciding with substantial regeneration (*Fu et al., 2012*). We recently identified interleukin-6 (IL-6) to be upregulated in the adult pituitary (in particular its stem cells) upon this local damage which activated the stem cells

(*Vennekens et al., 2021*). However, it is not clear what molecular mechanisms underlie the activation status of the stem cells in the neonatal gland.

Here, we set out to decode the stem cells' phenotype during the neonatal pituitary maturation stage starting from single-cell RNA-sequencing (scRNA-seq) interrogations, which uncovered an activation-designating hybrid epithelial–mesenchymal and WNT profile. We applied organoid and mouse models which supported and expanded the bioinformatic findings. Moreover, we found that the neonatal pituitary displayed high regeneration efficiency, more pronounced than in older mice (*Fu et al., 2012*; *Willems et al., 2016*), which is in line with its activated (stem cell) nature. Further decoding the molecular mechanisms underlying stem cell activation may open the door toward regenerative approaches for repairing harmed pituitary tissue, having serious endocrine consequences. In this prospect, our single-cell transcriptome database of neonatal (as well as adult) pituitary provides a highly valuable resource.

## Results

### Single-cell transcriptomics pictures prominent developmental activity in the neonatal maturing pituitary

To obtain a granular view on the cell type and activity landscape in the neonatal maturing pituitary, we performed scRNA-seq analysis of the major endocrine anterior pituitary (AP) from neonatal mouse (postnatal day 7, PD7), and contrasted the data with adult AP (*Vennekens et al., 2021*). After quality control and exclusion of doublets and dead and low-quality cells (*Figure 1—figure supplement 1A*; *Vennekens et al., 2021*), the neonatal AP cell transcriptomic data were integrated with our recent scRNA-seq dataset of adult AP (*Vennekens et al., 2021*) followed by filtering-out of ambient RNA (*Figure 1A*). Unsupervised clustering and annotation using established lineage markers (see *Cheung et al., 2018*; *Vennekens et al., 2021*) revealed all main endocrine cell types, as well as distinct clusters of pericytes and of stem, mesenchymal, immune, and endothelial cells (*Figure 1A, B*, *Figure 1—figure supplement 1B*). Of note, the neonatal data also revealed a small contaminating cluster with posterior lobe (PL) signature (i.e., *Nkx2-1*, *Rax* [*Cheung et al., 2018*] and pituicyte markers *Scn7a*, *Cldn10* [*Chen et al., 2020*]), in addition expressing the stem cell markers *Sox2* and *Sox9* (*Figure 1A, B*, *Figure 1—figure supplement 1B*), the latter in accordance with a previous report (*Cheung et al., 2018*). In general, sizable overlap between neonatal and adult cell clusters was observed (*Figure 1A*; visualization using Uniform Manifold Approximation and Projection [UMAP]). In particular, two stem cell subclusters (SC1 and SC2) were discerned in the neonatal AP as before also identified in the adult gland (*Vennekens et al., 2021*). These subclusters exhibit a diverging transcriptomic fingerprint, with SC1 displaying more prominent expression of keratins *Krt8* and *Krt18*, and SC2 of *Six1* and *Sox2* (*Figure 1—figure supplement 1C*), similar to the adult gland (*Vennekens et al., 2021*). Notwithstanding considerable overlap, clear differences emerged between neonatal and adult pituitary. First, the stem cell populations (both SC1 and SC2) are larger in neonatal than adult AP (three- to fourfold; *Figure 1A*), corroborated in situ by immunofluorescence analysis of the SOX2$^+$ cells (*Figure 1—figure supplement 1D*). Moreover, higher proliferative activity was observed in this SOX2$^+$ cell compartment (as assessed by Ki67 immunoanalysis; *Figure 1—figure supplement 1D*), as also found before (*Gremeaux et al., 2012*). In accordance, the scRNA-seq interrogation exposed a clear proliferative stem cell cluster (Prolif SC; as identified by coexpression of proliferation genes *Mki67*, *Pcna*, *Top2a*, *Mcm6*, with stem cell markers *Sox2*, *Krt8*, *Sox9*) in the neonatal AP, not discerned in the adult gland (*Figure 1A, B*, *Figure 1—figure supplement 1B*). Gene Ontology (GO) analysis using differentially expressed genes (DEGs) indeed revealed enrichment of cell cycle processes in the Prolif SC versus the aggregate SC1 and SC2 clusters (*Figure 1—figure supplement 2A*; *Figure 1—figure supplement 2—source data 1*).

Interestingly, proliferative subclusters were also perceived for other cell types, moreover exclusively in the neonatal gland. First, we discerned a proliferative *Pou1f1$^+$* (*Pit1$^+$*) cell group. *Pou1f1* represents the transcriptional regulator of hormone expression in somatotropes, lactotropes, and thyrotropes, and its expression also marks differentiation of progenitors within this so-called *Pou1f1$^+$* lineage (*Zhu et al., 2015*). In addition, we distinguished a proliferative corticotrope subcluster. Both findings were corroborated by DEG/GO and hormone/Ki67 immunostaining analyses (*Figure 1—figure supplement 2B*; *Figure 1—figure supplement 2—source data 2*). Strikingly, the neonatal AP also contains

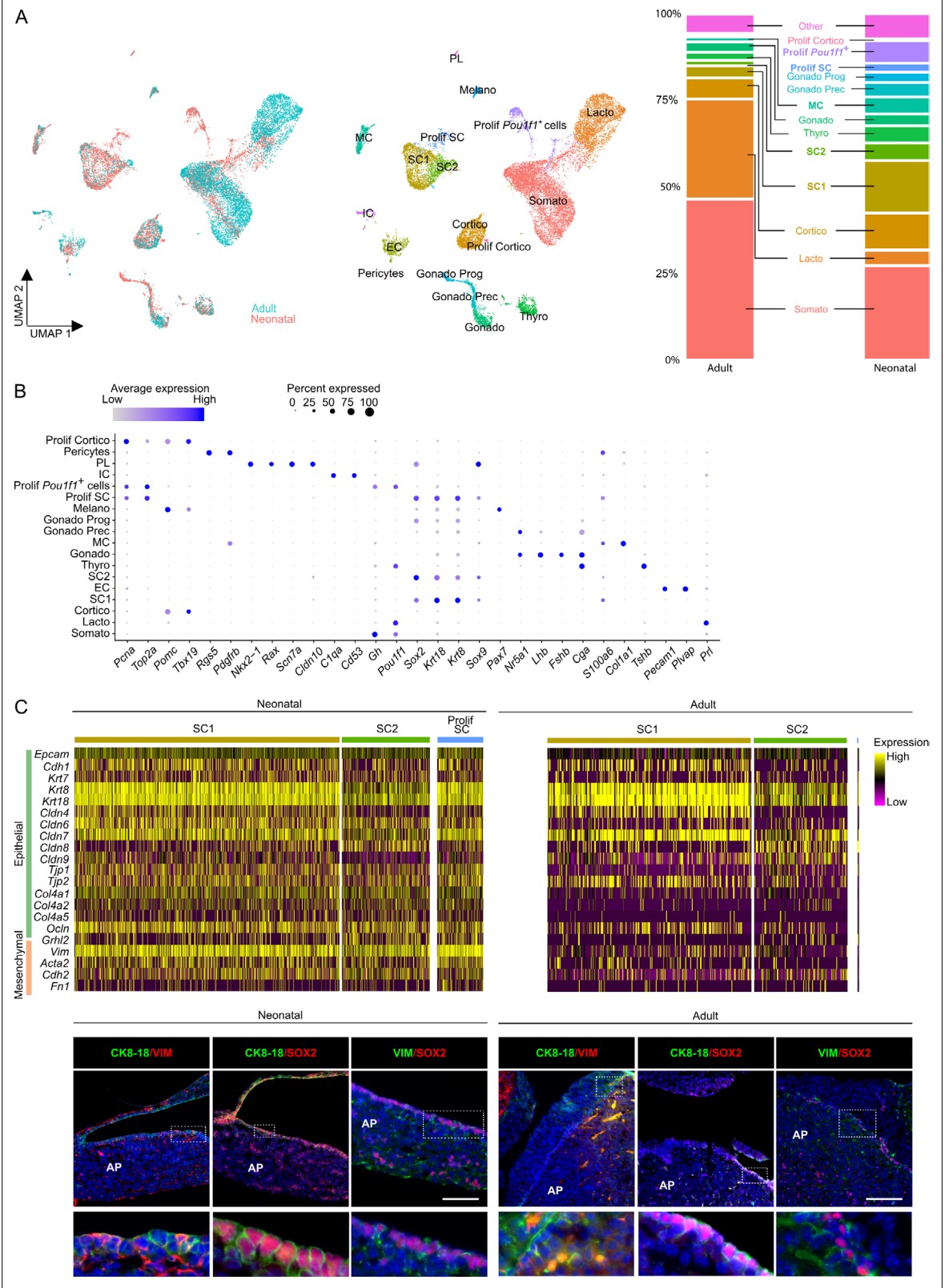

**Figure 1.** Single-cell transcriptomics of the neonatal maturing pituitary and its activated stem cell compartment. (**A**) *Left*: UMAP plot of neonatal and adult anterior pituitary (AP) combined. *Middle*: UMAP plot of the annotated cell clusters in the integrated AP samples (i.e., collective single-cell transcriptome datasets from neonatal and adult AP). Somato, somatotropes; Lacto, lactotropes; (Prolif) Cortico, (proliferating) corticotropes; Gonado, gonadotropes; Thyro, thyrotropes; Melano, melanotropes; SC1 and SC2, stem cell clusters 1 and 2; EC, endothelial cells; IC, immune cells;

*Figure 1 continued on next page*

*Figure 1 continued*

MC, mesenchymal cells; PL, posterior lobe cells; Gonado Prog, gonadotrope progenitor cells; Gonado Prec, gonadotrope precursor cells. *Right*: Bar plots showing proportions of each cell cluster at both ages. (**B**) Dot plot displaying percentage of cells (dot size) expressing indicated marker genes with average expression levels (color intensity; see scales on top) in the collective AP samples (i.e., adult and neonatal AP). (**C**) *Top*: Heatmap displaying scaled expression of selected epithelial and mesenchymal marker genes in the stem cell clusters SC1, SC2, and Prolif SC of neonatal and adult AP. *Bottom*: Immunofluorescence staining of CK8/18 (green), VIM (red/green), and SOX2 (red) in neonatal and adult AP. Nuclei are stained with Hoechst33342 (blue). Boxed area is magnified (scale bar, 100 µm).

The online version of this article includes the following source data and figure supplement(s) for figure 1:

**Figure supplement 1.** Single-cell transcriptomics of the neonatal maturing pituitary and its activated stem cell compartment.

**Figure supplement 2.** Single-cell transcriptomics of the neonatal maturing pituitary and its activated stem cell compartment.

**Figure supplement 2—source data 1.** Differentially expressed gene (DEG) and Gene Ontology (GO) analysis in Prolif SC *versus* SC1+SC2 of neonatal anterior pituitary (AP).

**Figure supplement 2—source data 2.** Differentially expressed gene (DEG) and Gene Ontology (GO) analysis in Prolif *Pou1f1*⁺ and Prolif Cortico cells *versus* Somato and Cortico, respectively, of neonatal anterior pituitary (AP).

**Figure supplement 2—source data 3.** Differentially expressed gene (DEG) and Gene Ontology (GO) analysis in Gonado Prog *versus* SC1+SC2+Prolif SC of neonatal anterior pituitary (AP).

gonadotrope progenitor and precursor subclusters as characterized by gradually increasing gonadotrope markers (*Nr5a1*, *Cga*, *Lhb*, *Fshb*, and *Gnrhr*) in parallel with declining stem cell factors (*Sox2*, *Krt8*, and *Sox9*) (*Figure 1A, B*, *Figure 1—figure supplement 1B*). The gonadotrope lineage arises last during embryonic development just before birth (*Stallings et al., 2016*), and some cells in the neonatal AP indeed coexpress SOX2 and the common gonadotropin subunit αGSU (*Figure 1—figure supplement 2C*). In further support of an endocrine progenitor phenotype (as opposed to a stem cell state), DEG analysis revealed upregulation of genes involved in hormone production (e.g., *Chga*, *Ascl1*, *Cga*, and *Insm1*) and enrichment of GO terms related to endocrine differentiation and secretion when compared to the stem cell clusters (*Figure 1—figure supplement 2C*; *Figure 1—figure supplement 2—source data 3*).

Taken together, scRNA-seq interrogation captured and embodied the prominent developmental activity that is taking place in the maturing neonatal gland, epitomized in the stem cell as well as endocrine lineage phenotypes. While the hormonal cell types (particularly somatotropes and lactotropes) rise in abundance toward adulthood, stem cells as well as supportive mesenchymal cells (MC) are more numerous in the neonatal gland (*Figure 1A*), suggesting that the latter populations play an active role in the critical development and growth of the gland at this acute period after birth, which becomes less prominent at mature age.

## Neonatal pituitary stem cells display a hybrid epithelial/mesenchymal phenotype

Using the scRNA-seq dataset, we assessed the epithelial character of the stem cell clusters. Surprisingly, the stem cell compartment of the neonatal AP displays a highly mixed epithelial/mesenchymal (E/M) phenotype, manifestly expressing both epithelial and mesenchymal markers, which is clearly faded in the adult gland (*Figure 1C*). In accordance, cells that coexpress the epithelial markers cytokeratin 8 and 18 (CK8/18) and the mesenchymal marker vimentin (VIM) were found in the marginal zone (MZ) SOX2⁺ stem cell niche of the neonatal pituitary, and were visibly less prominent in the adult gland (*Figure 1C*). Similarly, coexpression of SOX2 with VIM was more pronounced in the neonatal gland (*Figure 1C*). In other developing tissues, stem/progenitor cells also show a hybrid E/M character, indicative of their activation and participation in the tissues' developmental process (*Dong et al., 2018*). Recently, a hybrid E/M phenotype was also uncovered in the stem/progenitor cell cluster of the fetal human pituitary, with the mesenchymal aspect lowering along further maturation (*Zhang et al., 2020*). Moreover, epithelial-to-mesenchymal transition has been reported to play a role in mouse pituitary development through driving the exit of stem cells from the MZ toward the developing AP (*Pérez Millán et al., 2016*; *Yoshida et al., 2016*).

## Organoid culturing recapitulates the neonatal pituitary stem cell phenotype

To further explore the stem cells' nature of neonatal pituitary, we applied our recently established pituitary organoid model (*Cox et al., 2019*; *Vennekens et al., 2021*). We have shown that organoids developing from adult mouse AP originate from the SOX2+ stem cells and reflect their phenotype and activation state (e.g., higher organoid number following damage [*Cox et al., 2019*; *Vennekens et al., 2021*]), thus serving as a valuable pituitary stem cell biology research model and activation readout tool (*Cox et al., 2019*; *Vennekens et al., 2021*).

Dissociated AP cells from PD7 mice, embedded in Matrigel droplets (*Figure 2A*) and cultured in medium previously defined to grow organoids from adult AP (*Cox et al., 2019*) (referred to as pituitary organoid medium or PitOM, *Appendix 1—table 1*), were found to generate organoid structures, with a trend of higher number (although not significant; p = 0.18) than from adult AP (*Figure 2B*). Surprisingly, individual removal of several growth and signaling factors (i.e., fibroblast growth factors [FGF] 2/8/10, nicotinamide, sonic hedgehog [SHH], the transforming growth factor-β [TGFβ] inhibitor A83-01, the bone morphogenetic protein [BMP] inhibitor noggin and insulin-like growth factor 1 [IGF-1]) from the adult AP-geared PitOM resulted in a visible increase in organoid number developing from neonatal AP (*Figure 2—figure supplement 1A*), indicating that these PitOM-included factors are not essential (rather impedimental) for organoid outgrowth from the neonatal gland. Indeed, the neonatal pituitary stem cells show high expression of several of these factors, suggesting already prominent intrinsic pathway activity. Among others, *Fgf10* and *Igf1*, together with their receptors *Fgfr2* and *Igf1r*, and noggin (*Nog*) are found upregulated in the neonatal stem cell compartment (SC1+SC2+Prolif SC) as compared to the adult stem cells (*Figure 2—figure supplement 1B*). Moreover, this expression pattern is retained in the neonatal pituitary-derived organoids (*Figure 2—figure supplement 1B*). In contrast, cholera toxin (CT), p38 mitogen-activated protein kinase (MAPK) inhibitor (p38i; SB202190), the WNT-signaling amplifier R-spondin 1 (RSPO1) and epithelial growth factor (EGF) were found essential since organoid formation was largely abrogated in the absence of each one of these factors (*Figure 2—figure supplement 1A*). Together, this top-down screening by individually omitting PitOM factors led to an optimized medium for neonatal AP organoid development, further referred to as RSpECT (being the acronym of the essential core factors RSPO1, p38i, EGF and CT; *Appendix 1—table 1*).

The organoids, as before shown for adult AP (*Cox et al., 2019*), originated from the tissue-resident SOX2+ stem cells. Seeding AP from neonatal *Sox2*$^{eGFP/+}$ reporter mice (expressing enhanced green fluorescent protein [eGFP] in SOX2+ cells) generated only organoids that were fluorescent (eGFP+; *Figure 2—figure supplement 1C*). Furthermore, when *Sox2*$^{eGFP/+}$ AP cells were mixed with wildtype (WT) cells, organoids that developed were either eGFP+ or not (*Figure 2—figure supplement 1C*), thereby pointing to a clonal origin, as further supported by live-culture time-lapse imaging showing organoids growing from individual cells (*Figure 2—figure supplement 1D* and *Video 1*). The obtained organoids display a pituitary stemness phenotype expressing SOX2 as well as other known pituitary stem cell markers (such as E-cadherin, CK8/18, TACSTD2 (alias TROP2) [*Chen et al., 2009*; *Cox et al., 2019*; *Fauquier et al., 2008*; *Vennekens et al., 2021*]) (*Figure 2C*, *Videos 2 and 3*), while being devoid of hormone expression (*Figure 2—figure supplement 1E*). Using the optimized RSpECT medium, the number of organoids that developed from neonatal AP was significantly higher than from adult gland (for which RSpECT did not change the formed organoid number) (*Figure 2B*). Of note, the much larger number (~40-fold) can only partially be explained by the higher abundance of SOX2+ cells in the neonatal AP (~3- to 4-fold; see above). Indeed, normalized to the absolute number of SOX2+ cells seeded per well, the proportion of organoid-initiating SOX2+ cells is ~10-fold higher in neonatal than adult AP (*Figure 2—figure supplement 1F*). Thus, organoid formation efficiency reflects the higher intrinsic activation (or 'primed') status of neonatal pituitary stem cells per se.

Very recently, it has been reported that stem cells in early-postnatal (PD14) pituitary can function as autocrine and paracrine signaling center, among others stimulating proliferation within the own stem cell compartment (*Russell et al., 2021*). Interestingly, when co-cultured, neonatal (PD7) AP organoids (as developed from WT mice) elevate the outgrowth of organoids from adult AP (as established from ubiquitously tdTomato(tdT)-expressing *ROSA26*$^{mT/mG}$ mice) (*Figure 2D*), coinciding with increased proliferation of the adult AP organoid-constituting stem cells, showing a Ki67+ index reaching the one of neonatal AP-derived organoids (*Figure 2D*).

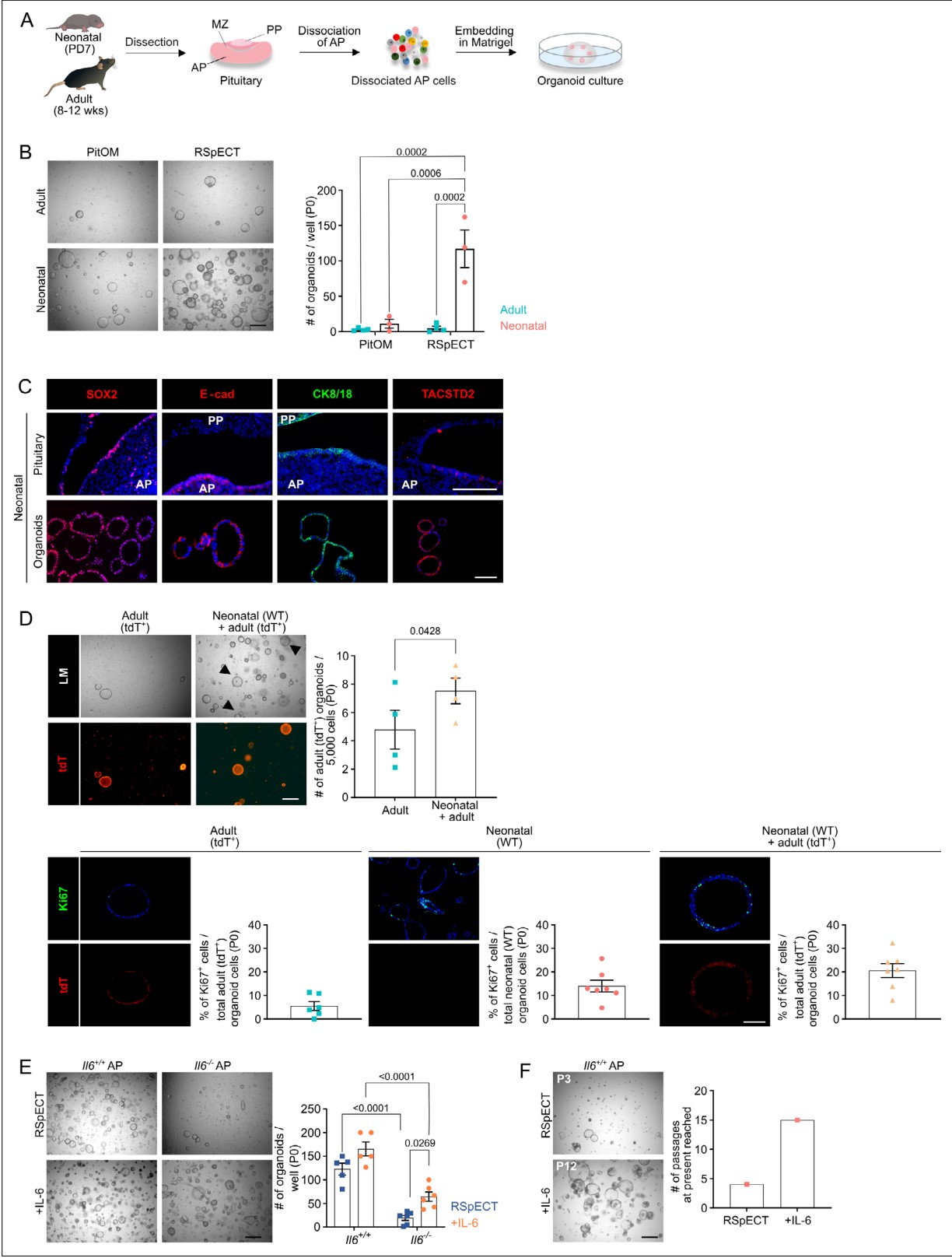

**Figure 2.** Organoids from neonatal pituitary recapitulate its stem cell phenotype. (**A**) Experimental schematic for organoid culturing. PD, postnatal day; wks, weeks; MZ, marginal zone; PP, posterior pituitary. Mouse icons obtained from BioRender. (**B**) Organoid formation efficiency (passage 0, P0) from adult and neonatal anterior pituitary (AP) cultured in PitOM or RSpECT medium. *Left*: Representative brightfield pictures of organoid cultures (scale bar, 500 μm). *Right*: Bar graph indicating number of organoids developed per well (mean ± standard error of the mean [SEM]). Data points

*Figure 2 continued on next page*

*Figure 2 continued*

represent biological replicates (*n* = 4 for Adult, *n* = 3 for Neonatal; two-way analysis of variance (ANOVA) with Tukey's multiple comparison test). (**C**) Immunofluorescence staining of SOX2, E-cadherin (E-cad), TACSTD2 (all red), and CK8/18 (green) in neonatal pituitary and derived organoids. Nuclei are labeled with Hoechst33342 (blue) (scale bar, 100 μm). (**D**) Organoid formation efficiency (P0) starting from AP cells of adult *ROSA26*$^{mT/mG}$ (tdTomato$^+$ (tdT)) mice, or from a 1:1 mixture of adult *ROSA26*$^{mT/mG}$ and neonatal wild type (WT) AP cells (P0). *Top left*: Light microscopic (LM) and epifluorescence (tdT) pictures. Arrowheads indicate tdT$^+$ organoids (scale bar, 500 μm). *Top right*: Bar graph indicating number of adult (tdT$^+$) organoids developing per 5000 cells in indicated cultures (mean ± SEM). Data points represent biological replicates (*n* = 4, paired *t*-test). *Bottom*: Immunofluorescence staining of Ki67 (green) and tdT (red) in AP organoids derived from adult *ROSA26*$^{mT/mG}$, neonatal WT or a 1:1 mixture of adult *ROSA26*$^{mT/mG}$ and neonatal WT AP cells. Nuclei are labeled with Hoechst33342 (blue) (scale bar, 100 μm). Bar graphs showing percentage of Ki67$^+$ cells in organoids as indicated (mean ± SEM). Data points represent individual organoids (*n* = 3). (**E**) Organoid formation efficiency from AP of neonatal *Il6*$^{+/+}$ and *Il6*$^{−/−}$ mice in RSpECT with or without interleukin-6 (IL-6) (P0). *Left*: Representative brightfield pictures of organoid cultures (scale bar, 500 μm). *Right*: Bar plot showing number of organoids developed per well in conditions as indicated (mean ± SEM). Data points represent biological replicates (*n* = 5 for *Il6*$^{+/+}$, *n* = 6 for *Il6*$^{−/−}$; two-way ANOVA with Tukey's multiple comparison test). (**F**) Neonatal AP organoid passageability with or without IL-6. *Left*: Representative brightfield images at indicated passage (scale bar, 500 μm). *Right*: Bars depicting the passage number reached at the end of this study.

The online version of this article includes the following figure supplement(s) for figure 2:

**Figure supplement 1.** Organoids from neonatal pituitary recapitulate its stem cell phenotype.

**Figure supplement 2.** Organoids from neonatal pituitary recapitulate its stem cell phenotype.

**Figure supplement 3.** Organoids from neonatal pituitary recapitulate its stem cell phenotype.

Taken together, organoids from neonatal AP reflect the stem cell compartment of the gland at this developmental stage regarding its activated and signaling center phenotype.

One factor that may be responsible for activating the stem cells in the neonatal pituitary is IL-6 which we recently uncovered as pituitary stem cell-activating factor in adult mouse (*Vennekens et al., 2021*). Intriguingly, gene expression of *Il6* is low in neonatal AP when compared to adult gland (*Figure 2—figure supplement 2A*). Similarly, gene-regulatory network (regulon) activity of the IL-6 signal-mediating JAK/STAT member *Stat3* is low as compared to adult gland (analysis using 'single-cell regulatory network inference' (SCENIC) [*Aibar et al., 2017*]) (*Figure 2—figure supplement 2A*). Notwithstanding, IL-6 is still produced, and beneficial, for AP organoid development. Indeed, a significantly smaller number of organoids grew from *Il6* knockout (*Il6*$^{−/−}$) versus WT neonatal AP, which was (partially) rescued by adding exogenous IL-6 (*Figure 2E*). Moreover, although IL-6 addition did not significantly increase the number of organoids formed from WT (*Il6*$^{+/+}$) neonatal AP (most likely because the stem cell activation status is already high) (*Figure 2E*), it strongly enhanced the passageability of the organoid cultures, from 4 to 5 passages without IL-6 to at least 15 passages (6 months of expansive culture) with the cytokine (*Figure 2F*),

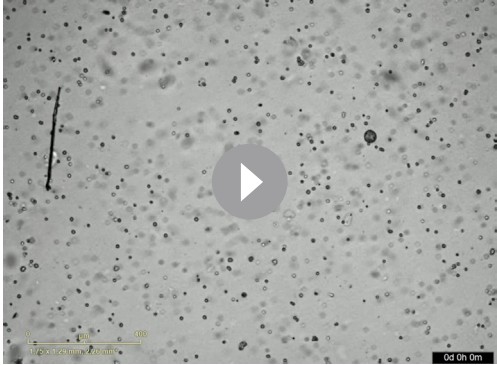

**Video 1.** Time-lapse video of neonatal anterior pituitary (AP) organoid formation. Video reconstruction of live time-lapse brightfield images captured by IncuCyte S3 (following the timepoints shown in the timetable on the video) of organoid culture after neonatal AP cell seeding (P0). Cultures were scanned automatically every 3 hr for 12 days.

https://elifesciences.org/articles/75742/figures#video1

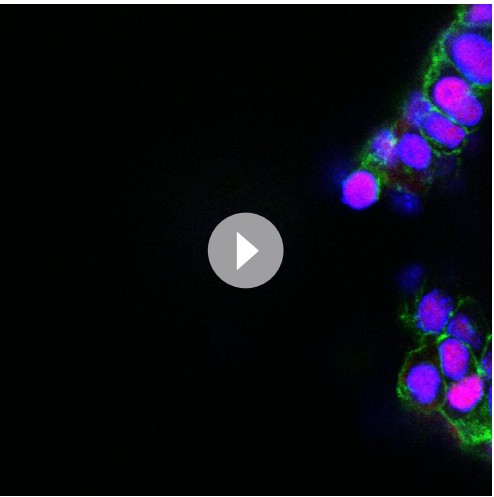

**Video 2.** 3D imaging of neonatal anterior pituitary (AP) organoids. Movie of *z*-stack through a neonatal AP organoid immunofluorescently stained for SOX2 (red) and E-cadherin (green). Hoechst33342 was used as nuclear stain (blue).

https://elifesciences.org/articles/75742/figures#video2

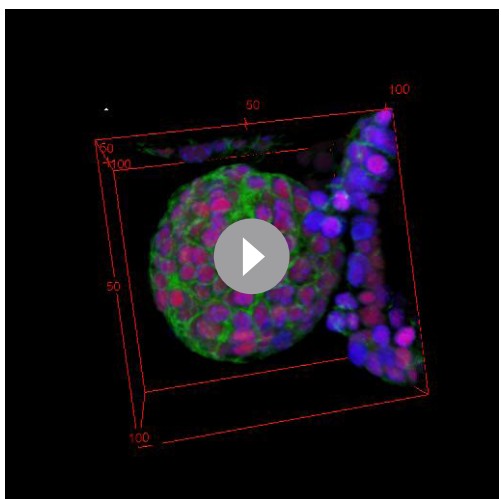

**Video 3.** 3D imaging of neonatal anterior pituitary (AP) organoids. 3D reconstruction of a neonatal AP organoid immunofluorescently stained for SOX2 (red) and E-cadherin (green). Hoechst33342 was used as nuclear stain (blue).

https://elifesciences.org/articles/75742/figures#video3

as also observed before for adult AP (*Vennekens et al., 2021*). Addition of IL-6 is considered to compensate for the decline in endogenous levels that is observed at passaging (*Figure 2—figure supplement 2B*). Of note, the organoids robustly retained their stemness phenotype during the long-term expansion, showing consistent stem cell marker expression (and hormone absence) between early (P2) and late (P8–11) passage (*Figure 2—figure supplement 2C*). Intriguingly, the in vivo lack of IL-6 does not visibly affect the SOX2$^+$ stem cell compartment and its proliferative phenotype (*Figure 2—figure supplement 3A*). Also, expression of other stem cell markers and of lineage progenitor and endocrine cell markers is not changed in the *Il6*$^{-/-}$ neonatal AP (*Figure 2—figure supplement 3B*). One possible explanation is that IL-6's function is in vivo taken over by other cytokines when IL-6 is absent from conception. Expression of the IL-6 family members *Il11* and leukemia-inhibitory factor (*Lif*), and of the related cytokine tumor necrosis factor-α (*Tnf*), was found slightly enhanced in the neonatal *Il6*$^{-/-}$ AP (*Figure 2—figure supplement 3C*). These cytokines, as well as the related IL-1β, did not significantly increase the number of organoids formed, but enhanced organoid expandability (*Figure 2—figure supplement 3D*), all similar to IL-6 (*Figure 2E, F*). The effects of TNFα and IL-1β seem to be mediated by IL-6 since these cytokines did not rescue its absence in organoid culturing from *Il6*$^{-/-}$ AP (*Figure 2—figure supplement 3E*), and increased the expression of IL-6 (as measured in WT AP organoids) (*Figure 2—figure supplement 3F*). In contrast, the IL-6 family members LIF and IL-11, known to act through the same gp130/JAK–STAT pathway (*Rose-John, 2018*), could substitute for IL-6 since increasing the number of developing organoids from *Il6*$^{-/-}$ AP as well as their expansion (passaging), while not affecting *Il6* expression in WT AP organoids (*Figure 2—figure supplement 3E, F*). An important underlying role of JAK–STAT signaling in organoid development as well as expansion is supported by the strong reduction of both processes when adding the STAT3 inhibitor STATTIC (*Figure 2—figure supplement 3G*). Taken together, the adult pituitary stem cell activator IL-6, although not highly expressed in neonatal AP stem cells, still advances organoid culturing from neonatal AP but appears not absolutely required for in situ stem cell maintenance and proliferative behavior in the neonatal gland, potentially due to redundancy of other IL-6 family cytokines.

## The neonatal pituitary stem cell compartment presents a pronounced WNT pathway

To continue the search for molecular mechanisms underlying the activated stem cell phenotype in the neonatal pituitary, we in further detail compared the neonatal and adult AP scRNA-seq datasets. DEG analysis revealed significantly increased expression of multiple WNT signaling-associated genes (e.g., *Frizzled* (*Fzd*)*2*, *Fzd3*, *Gsk3b*, *Ctnnb1*, and *Tnks*) in the neonatal versus adult stem cell clusters (*Figure 3A*; *Figure 3—source data 1A*). In analogy, GO analysis exposed that biological WNT pathway terms are enriched in the neonatal stem cell compartment (*Figure 3B*; *Figure 3—source data 1B*), and GSEA examination showed significant enrichment of WNT-associated hallmarks (*Figure 3—figure supplement 1A*). Regulon activity of the WNT downstream transcription factors *Tcf7l1* and *Tcf7l2* is most prominent in the stem cell clusters (SC1, SC2, and Prolif SC) when compared to the endocrine cell clusters (*Figure 3C*). Besides, high regulon activity is also present in the MC and EC clusters (*Figure 3C*). Clearly, regulon activity of these WNT transcription factors is higher in neonatal than adult gland (*Figure 3C*, violin plots). Furthermore, looking at upstream WNT pathway components, we found that ligands and receptors are overall higher expressed at neonatal age as

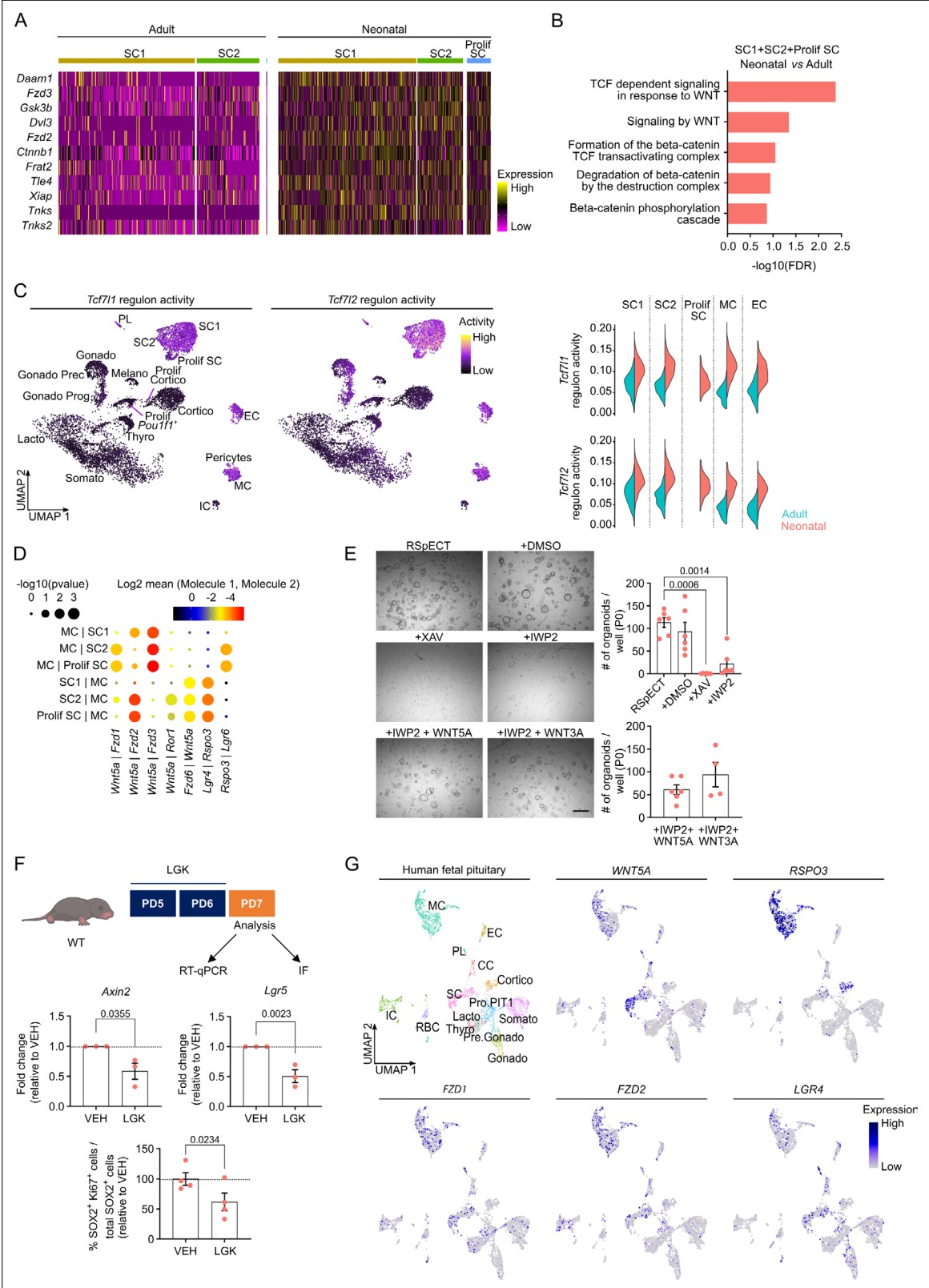

**Figure 3.** Neonatal pituitary stem cells show a pronounced WNT profile. (**A**) Heatmap displaying scaled expression of selected WNT-associated differentially expressed genes (DEGs) in the stem cell clusters SC1, SC2, and Prolif SC between adult and neonatal anterior pituitary (AP). (**B**) DEG-associated Gene Ontology (GO) terms linked with WNT signaling enriched in SC1, SC2, and Prolif SC of neonatal versus adult AP. (**C**) *Left*: *Tcf7l1* and *Tcf7l2* regulon activity projected on UMAP plot of neonatal AP, with indication of cell clusters. *Right*: Violin plots displaying regulon activity of *Tcf7l1*

*Figure 3 continued on next page*

*Figure 3 continued*

and *Tcf7l2* in the indicated clusters of adult and neonatal AP. (**D**) Dot plot displaying selected WNT-associated ligand–receptor interactions revealed by CellPhoneDB in neonatal SC1, SC2, Prolif SC, and MC clusters. p values are indicated by dot size, means of average expression of interacting molecule 1 in cluster 1 and interacting molecule 2 in cluster 2 are specified by color intensity (see scales on top). (**E**) Organoid development from neonatal AP cells, cultured and exposed to compounds as indicated (P0). *Left*: Representative brightfield pictures of organoid cultures (scale bar, 500 μm). *Right*: Bar graphs showing number of organoids formed per well under conditions as indicated (mean ± standard error of the mean [SEM]). Data points represent biological replicates (*n* = 6; one-way analysis of variance (ANOVA) with Tukey's multiple comparisons test). (**F**) *Top*: Schematic of in vivo treatment schedule and analysis. IF, immunofluorescence. Mouse icon obtained from BioRender. *Middle*: Bar plots depicting relative expression level of indicated genes in neonatal AP of mice treated as indicated (relative to vehicle [VEH], set as 1 [dashed line]) (mean ± SEM) (*n* = 3; unpaired *t*-test). *Bottom*: Bar graph showing percentage of SOX2$^+$Ki67$^+$ cells in SOX2$^+$ cell population in neonatal AP of mice treated as indicated (relative to VEH, set as 100% [dashed line]) (mean ± SEM). Data points represent biological replicates (*n* = 4; paired *t*-test). (**G**) UMAP plot of the annotated cell clusters in human fetal pituitary (*Zhang et al., 2020*) and projection of selected WNT-associated genes' expression. PL, posterior lobe (pituicyte) cells; CC, cell cycle cells; RBC, red blood cells; Pro.PIT1, progenitor cells of PIT1 lineage; Pre.Gonado, precursor cells of gonadotropes.

The online version of this article includes the following source data and figure supplement(s) for figure 3:

**Source data 1.** Differentially expressed gene (DEG) and Gene Ontology (GO) analysis in SC1+SC2+Prolif SC of neonatal *versus* adult anterior pituitary (AP).

**Figure supplement 1.** Neonatal pituitary stem cells show a pronounced WNT profile.

**Figure supplement 2.** Neonatal pituitary stem cells show a pronounced WNT profile.

analyzed in the whole AP (*Figure 3—figure supplement 1B*). Regarding the stem cell and MC clusters, particularly *Wnt5a* and *Fzd2* receptor are much higher expressed (both virtually absent in the adult AP) (*Figure 3—figure supplement 1C*). This age-related expression difference is even visible in whole AP gene expression analysis (RT-qPCR; *Figure 3—figure supplement 1C*). Taken together, the WNT pathway is more prominently present in the neonatal gland, which is proposed to contribute to the activated state of the stem cell compartment. A pronounced WNT profile has recently also been reported in the later PD14 mouse pituitary (*Russell et al., 2021*).

We have recently shown that the MC cluster is part of the formerly designated folliculo-stellate cell population in the (adult) AP (*Vennekens et al., 2021*), a heterogeneous cell group encompassing, among others, stem cells as well as paracrine-supportive cells (*Allaerts and Vankelecom, 2005*; *Chen et al., 2009*; *Fauquier et al., 2008*; *Vennekens et al., 2021*). Here, we applied CellPhoneDB (*Efremova et al., 2020*) on the neonatal scRNA-seq dataset to predict ligand–receptor interactions between MC and stem cells, thereby focusing on the prominently present WNT pathway. Multiple reciprocal interactions of *Wnt5a* with canonical (*Fzd*) and noncanonical (*Ror1*) receptors are projected (*Figure 3D*), with *Fzd1*, *Fzd3*, and *Fzd6* being the prevalent mediating receptors in the stem cells, and *Fzd2* both in the stem cells and MC (*Figure 3D*, *Figure 3—figure supplement 1D*). Within the RSPO/LGR-driven WNT amplification system, *Rspo3*, being predominantly expressed by MC cells, is forecasted to mainly bind to *Lgr4* and *Lgr6* receptors on the stem cells (*Figure 3D*, *Figure 3—figure supplement 1D*). RNAscope confirmed the pronounced expression of *Lgr4* and *Lgr6* in the *Sox2*$^+$ stem cells in situ, clearly different from *Rspo3* (*Figure 3—figure supplement 1D*).

To functionally validate the proposed stem cell-activating impact of WNT signaling in the neonatal gland, we first applied our in vitro organoid model. We observed that the stem cell WNT pathway components remained expressed in the organoids (*Figure 3—figure supplement 2A*). Addition of the WNT inhibitor XAV-939 (XAV; stabilizing the WNT-inhibitory protein AXIN2) at seeding (passage 0 [P0]) completely abolished organoid formation (*Figure 3E*). Exogenous administration of WNT ligands (WNT5A and WNT3A) to these cultures did not have an effect (*Figure 3—figure supplement 2B*), thereby validating the complete blockage of the intracellular (canonical) WNT-signaling path by the supplemented XAV. Addition of IWP2 (which blocks endogenous WNT ligand palmitoylation, as needed for secretion from the cell) to the seeded AP cells significantly reduced the number of organoids formed, which was substantially rescued by adding exogenous WNT5A and WNT3A (*Figure 3E*). Of note, both WNT inhibitors decreased the proliferative activity in the organoids (added to formed organoids at d7 of P0; *Figure 3—figure supplement 2C*). Together, these findings show that neonatal AP organoid growth, which reflects stem cell biology/activation, is dependent on (endogenous) WNT activity. Notably, RSPO1, which is typically used in organoid culturing (as we also did here and before, see *Appendix 1—table 1*; *Cox et al., 2019*; *Vennekens et al., 2021*), is only slightly expressed in the neonatal pituitary (*Figure 3—figure supplement 2D*) whereas *Rspo3* expression is prominent

(especially in the MC cluster; *Figure 3—figure supplement 1D*). This naturally abundant RSPO ligand was found capable to replace RSPO1 for efficient organoid formation (*Figure 3—figure supplement 2D*). Together, our findings advance the concept that the WNT pathway is an activator of pituitary stem cells. In support, addition of WNT5A or WNT3A to cultures from adult AP in which the stem cells are basically not activated (quiescent) (*Vennekens et al., 2021*) and the WNT pathway is much less prominent (*Figure 3A–D*, *Figure 3—figure supplement 1A, B*), increased stem cell proliferation in the organoids resulting in larger organoid structures (*Figure 3—figure supplement 2E*). Moreover, adding the WNT-signaling inhibitor Dickkopf 1 (DKK1) (which binds to the WNT co-receptors LRP5/6 thereby blocking WNT ligand action) to the co-culture of neonatal (WT) and adult (tdT$^+$) AP organoids resulted in decreased proliferative activity in the adult organoids (*Figure 3—figure supplement 2F*), strongly supporting that the proliferation-activating effect of the neonatal organoid (stem) cells on the adult organoid (stem) cells, when cultured together (see *Figure 2D*), is mediated, at least partly, by paracrine WNT activity.

To in vivo validate our organoid-based finding that the WNT pathway plays a role in stem cell activation in the neonatal pituitary, we treated neonatal (WT) pups with the WNT pathway (porcu-pine) inhibitor LGK-974 (LGK; *Figure 3F*). WNT target gene (*Axin2* and *Lgr5*) expression in the AP diminished following LGK administration, thereby verifying its activity and efficacy at the level of the pituitary (*Figure 3F*). Interestingly, the number of proliferating SOX2$^+$ cells decreased (*Figure 3F*), thus further supporting the involvement of WNT signaling in the activated phenotype of the neonatal AP stem cells.

Finally, as a first translation of our mouse-based pituitary findings to humans, we explored the recently published scRNA-seq dataset of fetal human pituitary (*Zhang et al., 2020*) regarding WNT component expression. First, we found that the mouse neonatal and fetal human pituitary showed significant overlap and high concordance in clustering outcome (*Figure 3—figure supplement 2G*). Of note, in utero development of humans is more extended than of mice (*Xue et al., 2013*), and the neonatal mouse stage thus rather corresponds to (late) embryonic stage in humans. Projection of above specified WNT-associated genes on the human fetal pituitary UMAP plot revealed a compa-rable expression pattern as in neonatal mouse AP, with, for instance, *WNT5A* being expressed in the SC and MC clusters, and *RSPO3* being most pronounced in the MC cluster (*Figure 3G*). Thus, the WNT pathway also appears highly present in fetal human pituitary where it may play comparable (e.g. stem-cell activating) roles. Of note, *IL6* is also expressed in the human fetal pituitary stem cells, moreover at similarly low levels as in the neonatal mouse pituitary (*Figure 3—figure supplement 2H*).

## The dynamic neonatal pituitary shows swift and complete regeneration after local damage

We have previously shown that somatotrope-ablation damage in the adult pituitary triggers acute prolifera-tive activation of the stem cell compartment and expression of GH in the SOX2$^+$ cells, and that the somato-trope population was eventually regenerated to 50–60% at 5–6 months after damage infliction (*Fu et al., 2012*). Here, we investigated whether the neonatal gland, housing a more activated ('primed') stem cell compartment, behaves differently regarding acute stem cell reaction and regeneration.

Three-day diphtheria toxin (DT) injection of neonatal (PD4) *Gh^{Cre/+}*;*ROSA26^{iDTR/+}* pups (further referred to as GHCre/iDTR mice, or 'damaged' condition) (*Figure 4A*) resulted in 50–60% ablation of GH$^+$ cells (*Figure 4B*) (as compared to 80–90% ablation at adult age [*Fu et al., 2012*]). The stem cell compartment did not visibly react (i.e., no increase in SOX2$^+$ cells and their proliferative index, neither proportion-wise nor in absolute cell numbers; *Figure 4C*, *Figure 4—figure supplement 1A*), which may be due to their already high activation status. Similarly, organoid formation did not significantly increase (*Figure 4—figure supplement 1B*). Fascinatingly, the somatotrope cell population was fully restored to normal numbers, moreover already achieved after 2 months (*Figure 4D*), meaning a more efficient regenerative capacity of the dynamic neonatal AP than the adult gland (*Fu et al., 2012*).

To start delving into the underlying mechanisms, we performed scRNA-seq analysis of damaged neonatal AP (the day after DT treatment (PD7); *Figure 4A*) and integrated the data with the control PD7 AP dataset as obtained above (*Figure 4E*, *Figure 4—figure supplement 1C*). First, in strong contrast to our recent findings in adult pituitary (*Vennekens et al., 2021*), *Il6* expression, being very low in neonatal pituitary (see above), was not upregulated in the stem cell and MC clusters following damage (*Figure 4—figure supplement 1D*), in line with a lack of extra stem cell activation (while IL-6

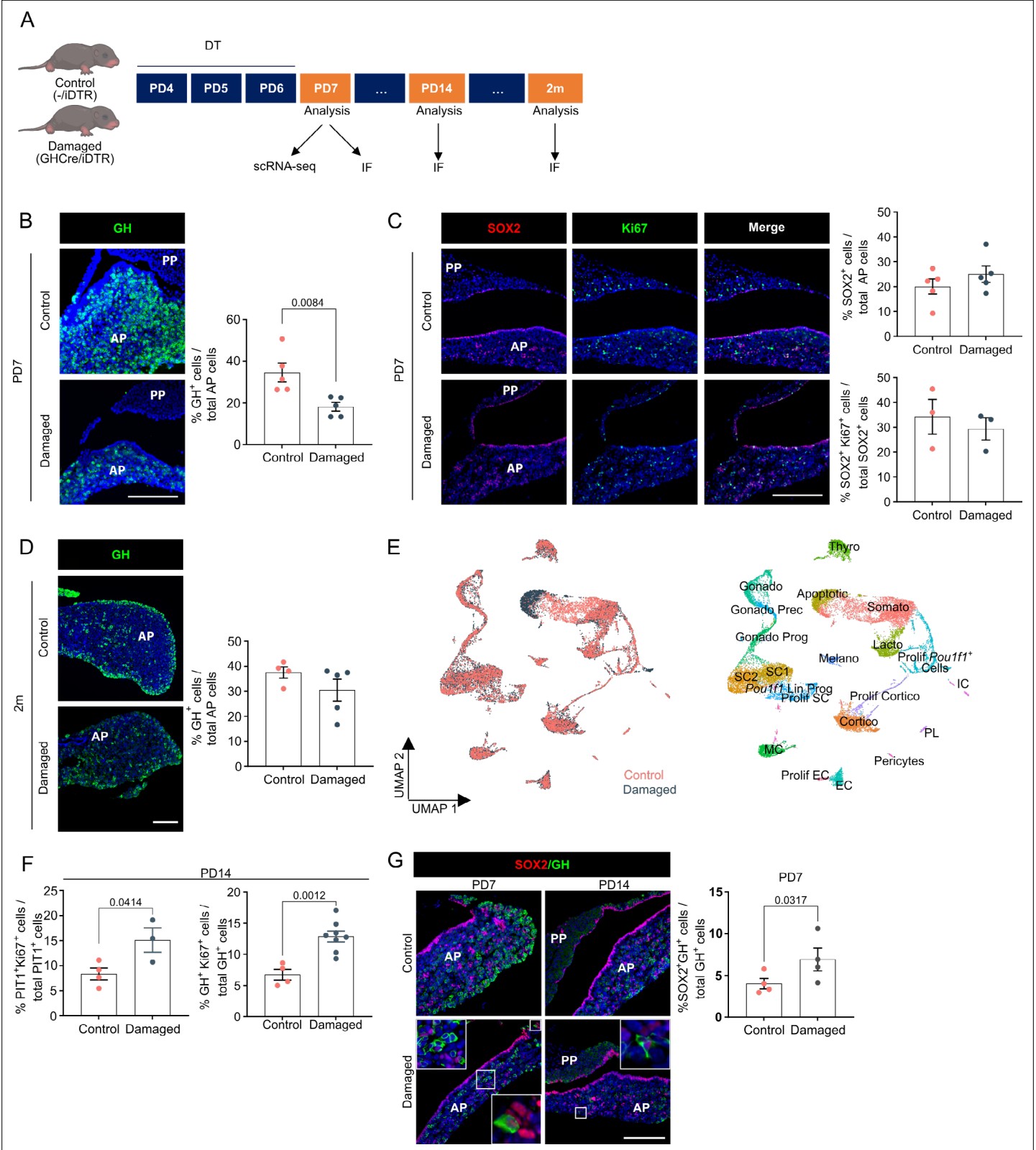

**Figure 4.** Neonatal pituitary's reaction to local damage and efficient regeneration. (**A**) Schematic of in vivo treatment schedule and analysis. DT, diphtheria toxin; m, months. Mouse icons obtained from BioRender. (**B**) Ablation of somatotropes (GH+ cells) in neonatal mouse anterior pituitary (AP). *Left*: Immunofluorescence staining of GH (green) in control and damaged pituitary following DT injection (PD7). Nuclei are labeled with Hoechst33342 (blue) (scale bar, 100 μm). *Right*: Bar graph showing proportion of GH+ cells in AP as indicated (mean ± standard error of the mean [SEM]). Data points

*Figure 4 continued on next page*

*Figure 4 continued*

represent biological replicates (*n* = 5; paired *t*-test). (**C**) SOX2⁺ stem cell reaction to damage. *Left*: Immunofluorescence staining of SOX2 (red) and Ki67 (green) in control and damaged pituitary following DT injection (PD7). Nuclei are labeled with Hoechst33342 (blue) (scale bar, 100 µm). *Right*: Bar graphs showing proportion of SOX2⁺ cells in AP as indicated, or of SOX2⁺Ki67⁺ cells in SOX2⁺ cell population (mean ± SEM). Data points represent biological replicates (*n* = 5 for % SOX2⁺ cells, *n* = 3 for % SOX2⁺Ki67⁺ cells). (**D**) Regeneration of somatotropes in neonatal mouse AP after their ablation. *Left*: Immunofluorescence staining of GH (green) in control and damaged pituitary 2 months after DT-induced ablation. Nuclei are labeled with Hoechst33342 (blue) (scale bar, 100 µm). *Right*: Bar graph showing proportion of GH⁺ cells in AP as indicated (mean ± SEM). Data points represent biological replicates (*n* = 4 for Control, *n* = 5 for Damaged). (**E**) *Left*: UMAP plot of control and damaged neonatal AP combined. *Right*: UMAP plot of annotated cell clusters in the integrated neonatal AP samples (i.e., collective single-cell transcriptome datasets from control and damaged AP). (**F**) Bar plots depicting proportion of PIT1⁺Ki67⁺ or GH⁺Ki67⁺ cells in PIT1⁺ or GH⁺ cell population, respectively, 1 week after DT-induced damage (PD14) (mean ± SEM). Data points represent biological replicates (*n* = 4 for Control, *n* = 3–8 for Damaged; unpaired *t*-test). (**G**) *Left*: Immunofluorescence staining of SOX2 (red) and GH (green) in control and damaged neonatal pituitary at indicated timepoints (PD7 and PD14) following DT-induced injury. Nuclei are stained with Hoechst33342 (blue). Boxed areas are magnified (scale bar, 100 µm). *Right*: Bar graph showing proportion of SOX2⁺GH⁺ cells in GH⁺ cell population following DT-induced damage (PD7) (mean ± SEM). Data points represent biological replicates (*n* = 4; paired *t*-test).

The online version of this article includes the following source data and figure supplement(s) for figure 4:

**Figure supplement 1.** Neonatal pituitary's reaction to local damage and efficient regeneration.

**Figure supplement 2.** Neonatal pituitary's reaction to local damage and efficient regeneration.

**Figure supplement 2—source data 1.** Differentially expressed gene (DEG) and Gene Ontology (GO) analysis in Prolif *Pou1f1*⁺ cells + Pou1f1 Lin Prog of damaged *versus* control neonatal anterior pituitary (AP).

**Figure supplement 2—source data 2.** Differentially expressed gene (DEG) and Gene Ontology (GO) analysis in SC1+SC2+Prolif SC of control versus damaged neonatal anterior pituitary (AP).

activates the stem cells in adult pituitary following damage [*Vennekens et al., 2021*]), together indicating that there is a striking difference in adult and neonatal pituitary (stem cell) reaction to damage. Moreover, also in line with the absence of additional stem cell activation, the WNT pathway appeared not to become extra fired, neither immediately upon damage (*Figure 4—figure supplement 1E*), nor at later timepoints during the regenerative period (i.e., one (PD14) and two (PD21) weeks after damage infliction), as assessed by *Lgr5* and *Axin2* target gene expression (*Figure 4—figure supplement 1E*). Intriguingly, a cluster of *Pou1f1* lineage progenitors (*Pou1f1* Lin Prog) became distinguishable in the aggregate (control + damaged) neonatal AP analysis (*Figure 4E*). DEG and GO examination of the two *Pou1f1* clusters together (Prolif *Pou1f1*⁺ cells and *Pou1f1* Lin Prog) revealed enriched cell proliferation genes and terms in damaged versus control condition (*Figure 4F*; *Figure 4—figure supplement 2A*; *Figure 4—figure supplement 2—source data 1*). In accordance, in situ proliferative activity of the PIT1⁺ cells was found increased upon damage (*Figure 4F*), as well as of already GH-expressing cells (*Figure 4F*), the latter not observed in adult gland (*Fu et al., 2012*). Finally, there is also an increase in SOX2⁺GH⁺ cells in damaged versus control gland (*Figure 4G*), suggesting rising stem cell differentiation. DEG and GO analysis of all SC clusters together revealed enriched NOTCH signaling terms in control versus damaged pituitary, or in others words, that NOTCH signaling is downregulated in the stem cells following the inflicted injury (*Figure 4—figure supplement 2B*; *Figure 4—figure supplement 2—source data 2*), which may be in line with the knowledge that NOTCH inhibition is needed for proper endocrine cell differentiation from stem/progenitor cells as observed during pituitary embryogenesis (*Cox et al., 2017*; *Zhu et al., 2006*). Taken together, restoration of the GH⁺ cell population in the neonatal pituitary may encompass several mechanisms including proliferative activation of the PIT1⁺ (progenitor) and derived (or existing) GH⁺ cell populations, and GH/somatotrope differentiation of SOX2⁺ stem cells, together embodying the active nature of the neonatal gland.

In conclusion, these data show that the dynamic neonatal gland more efficiently regenerates after injury than the adult gland, but that the already highly activated stem cell population does not show extra visible activation. Further insights into the underlying molecular mechanisms of neonatal AP regeneration will derive from still more extended scRNA-seq and functional explorations.

## Discussion

In our study, we in detail portray the activated nature of the neonatally maturing pituitary, in particular of its stem cells, using single-cell transcriptome profiling and functional in vitro (organoid) and in vivo (mouse) exploration. The stem cell compartment, shown to be expanded and proliferatively

activated in the neonatal gland, presents a pronounced WNT profile. Inhibiting WNT-signaling abrogated organoid formation and decreased proliferative activity of the neonatal pituitary stem cell compartment in vivo, thereby indicating the importance of the WNT pathway in the activated stem cell phenotype. This WNT signaling may act between the stem cells and supportive mesenchymal cells as projected by CellPhoneDB. Furthermore, the stem cells display a prominent hybrid E/M phenotype in the neonatal gland. A hybrid E/M character has recently also been found in the active stem/progenitor population of the fetal human pituitary (*Zhang et al., 2020*). More in general, a hybrid E/M nature is present in stem/progenitor cells of developing organs (such as lung, intestine, and liver), and underlies their active contribution to the tissues' development (*Dong et al., 2018*). Hence, the hybrid E/M phenotype of the neonatal pituitary stem cells likely embodies their active participation in the intense maturation process of the neonatal gland, as proposed during embryogenesis in human pituitary (*Zhang et al., 2020*). Along this line, we provide further indications that our findings in mouse are translatable to humans. Integrating our neonatal mouse AP scRNA-seq dataset with the recently published scRNA-seq data of fetal human pituitary (*Zhang et al., 2020*) revealed considerable overlap, including an analogous WNT landscape. A pronounced WNT profile has recently also been reported in the early-postnatal (2-week-old) mouse pituitary (*Russell et al., 2021*). Interestingly, this study showed a paracrine-regulatory role of these stem cells, stimulating neighboring committed progenitor cells (as well as stem cells) to proliferate and expand. We also show this paracrine-stimulatory nature of neonatal (1-week-old) pituitary stem cells, inducing proliferative activation of (adult) AP stem cell organoids. Moreover, organoid culturing recapitulated the activated stem cell phenotype of the neonatal AP showing higher outgrowth efficiency. Together, these findings again indicate the faithful reproduction of pituitary stem cell biology and activation by our organoid model.

We recently found that IL-6 is promptly upregulated in the adult pituitary, in particular in its stem cells, following transgenically inflicted damage, associated with proliferative stem cell activation (*Vennekens et al., 2021*). Intriguingly, *Il6* expression did not rise following damage in the neonatal gland. Together, these findings suggest that the pituitary reacts to damage differently according to developmental age. Along the same line, the dynamic neonatal pituitary more efficiently (faster and more extensively) regenerates than the adult pituitary following the local injury (*Fu et al., 2012*), likely due to the presence of activated ('primed') stem, committed progenitor and just differentiated endocrine cells that all appear to contribute by increased proliferation or differentiation (as supported by transcriptomic and in situ immunostaining analyses). Of note, the somewhat lower somatotrope ablation grade in neonatal pituitary may also partly contribute to the higher regeneration level. Interestingly, efficient regenerative capacity is also present in other mouse organs at neonatal age (such as heart and cochlea), while seriously declining or disappearing in adulthood (*Cox et al., 2014*; *Lam and Sadek, 2018*). Detailed unraveling of neonatal-pituitary regenerative mechanisms further needs profound studies, which may find ground in our existing and to be extended single-cell transcriptomic analyses.

*IL-6* expression in the neonatal gland is low (much lower than in adult pituitary), and its absence does not visibly affect the neonatally activated stem cell compartment. Either, IL-6 is not needed in the neonatal gland to induce or sustain the activated stem cell phenotype, or its lack has been taken over by cytokine family members such as LIF or IL-11. We provide support for the latter, and also propose that the JAK/STAT pathway may be the common denominator that lies at the basis of stem cell activation in the neonatal gland, hence also critical for organoid outgrowth (as we show with STATTIC and suggest with the *Il6*$^{-/-}$ mouse) and organoid expandability (as induced by the JAK/STAT-signaling factors IL-6, LIF, and IL-11). Adding IL-6 significantly prolonged the passageability of the organoids (as also found before for adult AP-derived organoids [*Vennekens et al., 2021*]), thereby compensating for its declining expression at culturing which may be due the stem cells' removal from the activating (micro-)environment and disappearance of stimulatory factors (such as IL-1β and TNFα which indeed stimulate IL-6 expression). Further research is now needed to unravel these hypotheses.

In conclusion, our study provides deeper insight into the activated phenotype of the neonatal pituitary stem cell compartment. Together with the scRNA-seq datasets and organoid models developed from adult and aging pituitary (*Cox et al., 2019*; *Vennekens et al., 2021*), we provide an arsenal of tools to compose a comprehensive view on pituitary stem cell biology, activation and role across key time points of life. Decoding stem cell activation will be needed and instrumental toward future aspirations of repairing and regenerating harmed pituitary tissue.

## Materials and methods

### Mice and in vivo treatments

Mice with C57BL/6 background were used for the experiments, which were approved by the KU Leuven Ethical Committee for Animal Experimentation (P153/2018). Mice were bred and kept in the animal housing facility of the KU Leuven under conditions of constant temperature, humidity, and day–night cycle, and had access to food and water ad libitum. For the neonatal pituitary experiments (biological replicates), male and female pups were pooled. For the adult pituitary experiments, biological replicates were performed with male or female mice, and results combined for statistical analyses.

*Sox2eGFP/+* (*Sox2tm1Lpev*) reporter mice contain the gene encoding for eGFP in the *Sox2* open reading frame, resulting in eGFP expression in SOX2-expressing cells (*Ellis et al., 2004*). Offspring was genotyped for the presence of the *eGFP* transgene by PCR using 5'-TACCCCGACCACATGAAGCA-3' as forward primer and 5'-TTCAGCTCGATGCGGTTCAC-3' as reverse primer.

*ROSA26mT/mG* mice (*Gt(ROSA)26Sortm4(ACTB-tdTomato,-EGFP)Luo*) contain, in the absence of Cre-mediated recombination, cell membrane-localized fluorescent tdTomato in all cells (*Muzumdar et al., 2007*).

*GhCre/+* mice (*Tg(Gh1-cre)bKnmn*) were crossed with *ROSA26iDTR/iDTR* animals (*Gt(ROSA)26Sortm1(H-BEGF)Awai*) to create *GhCre/+;ROSA26iDTR/+* offspring (i.e., heterozygous for both transgenes and abbreviated to GHCre/iDTR) as described in detail before (*Fu et al., 2012*). Offspring is genotyped for the presence of the *Cre* transgene by PCR using 5'-TGCCACGACCAAGTGACAGCAATG-3' as forward primer and 5'-ACCAGAGACGGAAATCCATCGCTC-3' as reverse primer, as previously described (*Fu et al., 2012*). GHCre/iDTR mice and *Cre*-negative control littermates (further referred to as −/iDTR) (PD4) were intraperitoneally (i.p.) injected with 4 ng DT (Merck, Darmstadt, Germany) per g bodyweight, twice a day (8 hr in between the injections) for 3 consecutive days. Pituitaries (damaged and control) were isolated and analyzed the following day (PD7), 1 week later (PD14), or 2 months later.

*Il6-/-* (*Il6tm1Kopf*) mice carry a targeted disruption of the *Il6* gene through replacement of the second exon by a neor cassette (*Kopf et al., 1994*). These mice were generously provided by Dr. P. Muñoz Cánoves (Cell Biology Unit, Pompeu Fabra University, Barcelona, Spain). Offspring was genotyped for the presence of the neor cassette and the WT *Il6* gene by PCR using 5'-TTCCATCCAGTTGCCTTCTT GG-3' as common forward primer, 5'-TTCTCATTTCCACGATTTCCCAG-3' as WT reverse primer, and 5'-CCGGAGAACCTGCGTGCAATCC-3' as mutant reverse primer.

WT (C57BL/6) neonatal mice (PD5) were treated twice with 5 µg LGK-974 (Biogems, Westlake Village, CA) per g bodyweight or vehicle (corn oil, Merck) trough oral gavage, for 2 consecutive days. Pituitaries were isolated and analyzed the following day (PD7).

### scRNA-seq analysis

The AP of neonatal (PD7) mice was isolated and dispersed into single cells using trypsin (Thermo Fisher Scientific, Waltham, MA), all as previously described (*Denef et al., 1978*; *Van der Schueren et al., 1982*). The eventual cell suspension was then subjected to scRNA-seq analyses (two biological replicates). Cells were loaded on a 10× Genomics cartridge according to the manufacturer's instructions based on 10× Genomics' GemCode Technology (10× Genomics, Pleasanton, CA). Barcoded scRNA-seq libraries were prepared with the Chromium Single-cell 3' v2 Chemistry Library Kit, Gel Bead & Multiplex Kit and Chip Kit (10× Genomics). The libraries were sequenced on an Illumina NextSeq and NovaSeq6000. Data are accessible from ArrayExpress database (accession number E-MTAB-11337). Raw sequencing reads were demultiplexed, mapped to the mouse reference genome (mm10) and gene expression matrices were generated using CellRanger (v3; 10× Genomics). Downstream analysis was performed in R (v.3.6.1) using Seurat (v.3.1.3) (*Butler et al., 2018*). First, low-quality/dead cells and potential doublets (i.e., with less than 750 genes or more than 8000 genes and more than 17.5% mitochondrial RNA; see cutoffs in *Figure 1— figure supplement 1A*) were removed, resulting in a total of 21,419 good-quality single cells (i.e., 9618 from undamaged AP and 11,801 from damaged AP) for downstream analyses. To allow integrated, comparative examination of undamaged and damaged AP samples, the standard Seurat v3 integration workflow was followed (*Stuart et al., 2019*). In short, after normalization and identification of variable features for each sample, integration anchors were identified using the FindIntegrationAnchors function with default parameters and dims = 1:20, and data were integrated across all features. Following integration, expression levels were scaled, centered and subjected to principal component analysis (PCA). The top 20 PCs were selected and used for UMAP dimensionality reduction (*McInnes et al., 2018*). Clusters were identified with the Find-Clusters function by use of the shared nearest neighbor modularity optimization with a clustering resolution

set to 1. This analysis resulted in the identification of 33 distinct clusters, which were annotated based on canonical (AP) cell markers and on previous mouse pituitary scRNA-seq reports (*Cheung et al., 2018*; *Ho et al., 2020*; *Mayran et al., 2019*; *Vennekens et al., 2021*). Background (ambient) RNA was removed using SoupX (v.1.4.5) with default parameters (*Ho et al., 2020*). The global contamination (ambient RNA) fraction was estimated at 1.4%, well within the common range of 0–10%. The SoupX-filtered expression matrices were then loaded into Seurat, and processed using standard preprocessing (normalization, variable feature selection and scaling) to enable further downstream analyses.

To integrate our neonatal pituitary dataset with our previously published adult pituitary dataset (*Vennekens et al., 2021*), we applied Seurat's reference-based integration approach, for which the 'adult' condition was used as reference dataset when applying the FindIntegrationAnchors function (as described above [*Stuart et al., 2019*]). The top 30 PCs were selected and used for UMAP dimensionality reduction (*McInnes et al., 2018*). Clusters were identified with the FindClusters function with a clustering resolution set to 1.6. This analysis resulted in the identification of 37 distinct clusters, which were annotated.

Differential gene expression analysis was performed using the FindMarkers function with default parameters. GO analysis of biological processes was executed on significant DEGs FDR ≤0.05 and logFC ≥0.25 using Reactome overrepresentation analysis v3.7 and GOrilla (*Eden et al., 2009*; *Fabregat et al., 2018*). Gene-set enrichment analysis (GSEA; v.4.1.0) was performed using normalized expression data (*Mootha et al., 2003*; *Subramanian et al., 2005*). Gene sets (hallmarks) tested were obtained from the Molecular Signatures Database (MSigDB; v.7.2) (*Liberzon et al., 2015*; *Subramanian et al., 2005*), and converted to mouse gene signatures using MGI batch query (http://www.informatics.jax.org/batch).

Gene regulatory networks (regulons) were determined in our integrated pituitary dataset using pySCENIC, that is, SCENIC (v.0.9.15 [*Aibar et al., 2017*]) in Python (v.3.6.9). Raw expression data were normalized by dividing feature counts of each cell by the total counts for that same cell and multiplying by 10,000 followed by log1p transformation. Next, co-expression modules were generated using GRNboost2 algorithm (v.0.1.3) (*Moerman et al., 2019*). Subsequently, gene regulatory networks were inferred using pySCENIC (with default parameters and mm10__refseqr80__10 kb_up_and_down_tss. mc9nr and mm10__refseqr80__500 bp_up_and_100 bp_down_tss.mc9nr motif collections) resulting in the matrix of AUCell values that represent the activity of each regulon in each cell. The AUCell matrix was imported into Seurat for further downstream analysis, after which it was integrated using the default integration method, as described above. To generate regulon-based UMAP plots, the integrated AUCell matrix was scaled, centered, and subjected to PCA analysis and the top 20 PCs were selected for UMAP representation.

For integration of our dataset with the recently published human fetal pituitary scRNA-seq dataset (*Zhang et al., 2020*), the standard Seurat v3 workflow was used as described above. Following integration, expression levels were scaled, centered, and subjected to PCA. The top 30 PCs were selected and employed for UMAP dimensionality reduction.

Interactions between pairwise cell clusters were inferred by CellPhoneDB v.2.1.5, which includes a public repository of curated ligands, receptors, and their interactions (*Efremova et al., 2020*). We ran the Cell-PhoneDB framework using a statistical method and detected ligand–receptor pairs that were expressed in more than 20% of cells. Selected significant ligand–receptor pairs (p-value ≤0.05 and mean value ≥0.5) are shown.

## Organoid culture and treatment

Organoids were developed and cultured as in detail described before (*Cox et al., 2019*; *Laporte et al., 2022*). In short, AP cells were plated at a density of 10,000 cells per 30 μl drop of growth factor-reduced Matrigel (Corning, New York, NY) mixed with serum-free defined medium (SFDM; Thermo Fisher Scientific) in a 70:30 ratio (*Figure 2A*). For all cultures, PitOM or RSpECT (*Appendix 1— table 1*) was used unless otherwise stated. At seeding of the primary AP cells and at replating of the organoid fragments (i.e., passaging), ROCK inhibitor (Y-27632; 10 μM; Merck) was added to the medium. Organoid cultures were passaged every 10–14 days; the organoids were incubated with TrypLE Express (Thermo Fisher Scientific) and mechanically dispersed until organoid fragments were obtained which were reseeded in Matrigel drops as above.

To explore their effect on organoid culturing, IL-6 (20 ng/ml; Peprotech, London, UK), IL-1β (10 ng/ml; Peprotech), TNFα (20 ng/ml; Peprotech), LIF (25 ng/ml; Peprotech), IL-11 (25 ng/ml; Peprotech) or STATTIC (20 μM; Merck) was added to the medium.

To assess WNT pathway involvement in organoid culturing, IWP2 (4 µM; Merck), XAV-939 (10 µM; Merck), WNT3A (200 ng/ml; R&D systems, Minneapolis, MN), WNT5A (100 ng/ml; AMSBIO, Cambridge, MA), or RSPO3 (200 ng/ml; Peprotech) were supplemented to the medium.

Brightfield and fluorescence pictures of organoid cultures were recorded using an Axiovert 40 CFL microscope (Zeiss, Oberkochen, Germany). Organoid-forming efficiency was determined by quantifying the number of clearly developed organoids (≥100 µm) in whole organoid culture drops using Fiji (https://imagej.net/Fiji; *Schindelin et al., 2012*).

## Histochemical and immunostaining analysis

Pituitary and organoids were fixed in 4% paraformaldehyde (PFA; Merck) and embedded in paraffin using the Excelsior ES Tissue Processor (Thermo Fisher Scientific). Sections were subjected to immunofluorescence staining as described earlier (*Cox et al., 2019*). Antigen retrieval in citrate buffer (Merck) was followed by permeabilization with Triton X-100 (Merck) and blocking with donkey serum (Merck). Following incubation with primary and secondary antibodies (Appendix 1—key resources table), sections were covered with ProLong Gold (Thermo Fisher Scientific) after nuclei counterstaining with Hoechst33342 (Merck).

To quantify $SOX2^+$, $Ki67^+$, $PIT1^+$, and $GH^+$ cells, dissociated AP cells were spun onto SuperFrost glass slides (Thermo Fisher Scientific) and the cytospin samples immunostained as described before (*Fu et al., 2012*). Proportions of immunoreactive cells were counted using Fiji software as previously described (*Vennekens et al., 2021*).

Images were recorded using a Leica DM5500 upright epifluorescence microscope (Leica Microsystems, Wetzlar, Germany) accessible through the Imaging Core (VIB, KU Leuven) and converted to pictures for figures with Fiji imaging software.

## Gene expression analysis

Total RNA was isolated using the RNeasy Micro kit (Qiagen, Hilden, Germany) and reverse-transcribed (RT) with Superscript III First-Strand Synthesis Supermix (Invitrogen, Waltham, MA). SYBR Green-based quantitative 'real-time' PCR (RT-qPCR) was performed, using specific forward and reverse primers (*Appendix 1—table 2*), as described before (*Cox et al., 2019*). β-Actin (*Actb*), displaying stable expression levels among the conditions tested, was used as housekeeping gene for normalization. Normalized gene expression levels are shown as bar graphs of dCt values (Ct target – Ct housekeeping gene), or gene expression levels were compared between sample and reference as relative expression ratio (fold change) using the formula $2^{-(dCt\ sample\ -\ dCt\ reference)}$.

## Time-lapse recording of organoid development and growth

AP cells were seeded at a density of 1000 cells in 5 µl Matrigel/SFDM drops in 96-well plates (Corning). Brightfield time-lapse images were recorded with the IncuCyte S3 (Sartorius, Göttingen, Germany) every 3 hr for 12 days. Time-lapse videos were generated with the IncuCyte software using 10 frames per second.

## 3D imaging of cleared organoids

Whole organoids were immunofluorescently stained and imaged as described (*Dekkers et al., 2019*). In short, organoids were removed from the Matrigel droplet and fixed in 4% PFA. Permeabilization and blocking were performed with Triton X-100 and bovine serum albumin (Serva, Heidelberg, Germany) prior to sequential incubation with primary and secondary antibodies (Appendix 1-key resources table). Clearing was achieved by incubating the organoids in a fructose-glycerol solution (Merck; Thermo Fisher Scientific). Samples were mounted and images recorded using a Zeiss LSM 780 – SP Mai Tai HP DS accessible through the Cell and Tissue Imaging Cluster (CIC; KU Leuven). Acquired z-stacks were imported into Fiji image analysis software and 3D reconstructions made using the 3D viewer plugin (*Pietzsch et al., 2015*).

## RNAscope in situ hybridization

Whole pituitary was fixed in 4% PFA for 24 hr at room temperature and then paraffin-embedded as described above. Five-µm sections were subjected to in situ hybridization with the RNAscope Multiplex Fluorescent Reagent Kit v2 (Advanced Cell Diagnostics, Newark, CA) following the manufacturer's instructions. Differently labeled RNAscope probes (Advanced Cell Diagnostics) were used for mouse *Sox2* (401041-C3), *Rspo3* (483781-C2), *Lgr4* (318321-C2), and *Lgr6* (404961-C2). Sections were

counterstained with DAPI, mounted with ProlongGold, and analyzed with the Zeiss LSM 780 – SP Mai Tai HP DS. Recorded images were converted to pictures using Fiji.

## Electrochemiluminescent measurement of IL-6

IL-6 protein levels were measured with the sensitive electrochemiluminescent 'Meso Scale Discovery' (MSD) V-PLEX mouse IL-6 kit (MSD; Rockville, MD), according to the manufacturer's protocol, in organoid culture supernatant which was collected and centrifuged for 10 min at 1500 rpm (4°C). Plates were run on the MESO QuickPlex SQ 120 reader and data were analyzed using the MSD discovery workbench software (v4.0.12).

## Acknowledgements

We thank Y Van Goethem and V Vanslembrouck for valuable technical help. We are also grateful to the Imaging Core (VIB, KU Leuven) and the CIC (KU Leuven) for use of microscopes and the Center for Brain & Disease Research (CBD) Histology unit (VIB, KU Leuven) for use of histology equipment. We are indebted to Dr Pura Muñoz-Cánoves (Universitat Pompeu Fabra, Barcelona, Spain) for generously providing the *Il6*$^{-/-}$ mice. We thank Thomas Van Brussel, Rogier Schepers, and Bram Boeckx (D L's group, KU Leuven) for technical and bioinformatical support in scRNA-seq experiments. The computational resources used for scRNA-seq analysis were provided by the 'Vlaams Supercomputer Centrum', managed by the Fund for Scientific Research (FWO)—Flanders. We also thank the FACS core (KU Leuven) for training and use of the MESO QuickPlex SQ 120 reader. Finally, we recognize the Laboratory of Virology and Chemotherapy (Rega Institute; Dr D Daelemans) for experimental help and use of the IncuCyte S3. Funding: This work was supported by grants from the KU Leuven Research Fund and from the Fund for Scientific Research (FWO)—Flanders. E L (11A3320N), A V (1141717 N), C N (1S14218N), and B C (11W9215N) are supported by a PhD Fellowship from the FWO/FWO-SB. Use of the Zeiss LSM 780 – SP Mai Tai HP DS is supported by Hercules AKUL/11/37 and FWO G.0929.15 funding to Dr P Vanden Berghe (CIC, KU Leuven). The funders had no role in study design, data collection, and interpretation, or the decision to submit the work for publication.

## Additional information

### Funding

| Funder | Grant reference number | Author |
| --- | --- | --- |
| Fonds Wetenschappelijk Onderzoek | 11A3320N | Emma Laporte |
| Fonds Wetenschappelijk Onderzoek | 1141717N | Annelies Vennekens |
| Fonds Wetenschappelijk Onderzoek | 1S14218N | Charlotte Nys |
| Fonds Wetenschappelijk Onderzoek | 11W9215N | Benoit Cox |
| KU Leuven | | Hugo Vankelecom |

The funders had no role in study design, data collection, and interpretation, or the decision to submit the work for publication.

### Author contributions

Emma Laporte, Conceptualization, Data curation, Formal analysis, Investigation, Methodology, Validation, Visualization, Writing - original draft, Writing - review and editing; Florian Hermans, Data curation, Formal analysis, Investigation, Methodology, Software; Silke De Vriendt, Formal analysis, Investigation, Methodology, Validation; Annelies Vennekens, Investigation, Validation; Diether Lambrechts, Data curation, Formal analysis, Investigation, Methodology, Resources, Software, Validation; Charlotte Nys, Investigation; Benoit Cox, Conceptualization, Investigation, Methodology;

Hugo Vankelecom, Conceptualization, Funding acquisition, Methodology, Project administration, Resources, Supervision, Writing - original draft, Writing - review and editing

**Author ORCIDs**
Emma Laporte https://orcid.org/0000-0003-0799-3116
Florian Hermans https://orcid.org/0000-0002-2321-3995
Silke De Vriendt https://orcid.org/0000-0002-4022-2869
Benoit Cox https://orcid.org/0000-0002-3139-268X
Hugo Vankelecom https://orcid.org/0000-0002-2251-7284

**Ethics**
Mice with C57BL/6 background were used for the experiments, which were approved by the KU Leuven Ethical Committee for Animal Experimentation (P153/2018).

**Decision letter and Author response**
Decision letter https://doi.org/10.7554/eLife.75742.sa1
Author response https://doi.org/10.7554/eLife.75742.sa2

## Additional files

**Supplementary files**
• Transparent reporting form

**Data availability**
scRNA-seq data have been deposited in ArrayExpress (E-MTAB-11337). All other study data are included in the article and/or supporting information.

The following dataset was generated:

| Author(s) | Year | Dataset title | Dataset URL | Database and Identifier |
|---|---|---|---|---|
| Hermans F, Laporte E, Vankelecom H | 2022 | Single-cell transcriptomics of neonatal mouse anterior pituitary in steady-state conditions and after transgenically inflicted local damage | https://www.ebi.ac.uk/biostudies/arrayexpress/studies/E-MTAB-11337 | ArrayExpress, E-MTAB-11337 |

The following previously published datasets were used:

| Author(s) | Year | Dataset title | Dataset URL | Database and Identifier |
|---|---|---|---|---|
| Hermans F, Vennekens A, Laporte E, Vankelecom H | 2021 | Single-cell transcriptional profiling of young-adult and aging (middle-aged) mouse anterior pituitary in steady-state conditions and after transgenically inflicted local damage | https://www.ebi.ac.uk/biostudies/arrayexpress/studies/E-MTAB-10021 | ArrayExpress, E-MTAB-10021 |
| Zhang S, Cui Y, Wen L, Qiao J, Tang F | 2020 | Single-cell Transcriptomics Reveals the Divergent Developmental Lineage Trajectories during Human Pituitary Development | https://www.ncbi.nlm.nih.gov/geo/query/acc.cgi?acc=GSE142653 | NCBI Gene Expression Omnibus, GSE142653 |

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

## Appendix 1

**Appendix 1—table 1.** Overview of medium components in PitOM and RSpECT, used for organoid culturing.

| Pituitary organoid medium (PitOM) | | | |
|---|---|---|---|
| **Component** | **Concentration** | **Catalogue number** | **Company** |
| **SFDM** | | | Thermo Fisher Scientific |
| **B27** | 1× | 12587010 | Thermo Fisher Scientific |
| L-Glutamine | 2 mM | 25030081 | Thermo Fisher Scientific |
| **N2** | 1× | 17502048 | Thermo Fisher Scientific |
| *N*-Acetyl-cysteine | 1.25 mM | A7250 | Sigma-Aldrich |
| **EGF** | 50 ng/ml | 236-EG | R&D systems |
| SB202190 (=p38 i) | 10 µM | S7067 | Sigma-Aldrich |
| Cholera toxin | 100 ng/ml | C8052 | Sigma-Aldrich |
| **RSPO1** | 200 ng/ml | 120-38 | Peprotech |
| Nicotinamide | 10 mM | N0636 | Sigma-Aldrich |
| bFGF (=FGF2) | 20 ng/ml | 234-FSE | R&D systems |
| IGF1 | 100 ng/ml | 100-11 | Peprotech |
| FGF8 | 200 ng/ml | 100-25 | Peprotech |
| FGF10 | 100 ng/ml | 100-26 | Peprotech |
| A83-01 | 0.5 µM | SML0788 | Sigma-Aldrich |
| SHH | 100 ng/ml | 464-SH | R&D systems |
| Noggin | 100 ng/ml | 120-10C | R&D systems |

**Bold**: components of **RSpECT** medium.

**Appendix 1—table 2.** Overview of primers, used for RT-qPCR.

| Primer sequences | | |
|---|---|---|
| **Target** | **Forward (5′ > 3′)** | **Reverse (5′ > 3′)** |
| *Actb* | GCTGAGAGGGAAATCGTGCGTG | CCAGGGAGGAAGAGGATGCGG |
| *Axin2* | CGACCCAGTCAATCCTTATCAC | GGGACTCCATCTACGCTACTG |
| *Cdh1* | AGAAGATCACGTATCGGATTTGG | TTCTTCACATGCTCAGCGTC |
| *Cga* | CAAGCTAGGAGCCCCCATCTA | CACTCTGGCATTTCCCATTACT |
| *Cntf* | TCTGTAGCCGCTCTATCTGG | GGTACACCATCCACTGAGTCAA |
| *Ct1* | AAGATCCGCCAGACACACAACC | TGAGAAGCCCGGCAGCCCAAA |
| *Fgf10* | ATCACCTCCAAGGAGATGTCCG | CGGCAACAACTCCGATTTCCAC |
| *Fgfr2* | GTCTCCGAGTATGAGTTGCCAG | CCACTGCTTCAGCCATGACTAC |
| *Fzd1* | CAGCAGTACAACGGCGAAC | GTCCTCCTGATTCGTGTGGC |
| *Fzd2* | AAGAATGACCTCCTTTCCCT | CACAACCCAATCCTACAAACAG |
| *Fzd3* | CCCTATTTGTATGGAATATGGACG | TCATCACAATCTGGAAACCTACTG |
| *Fzd6* | TCTTCCCTAACCTGATGGGTC | ACAATTTCCGACAGGGTAGAAC |
| *Gata2* | AAGGATGGCGTCAAGTACCAA | TATCGGGTGGTGTGTTGCAG |
| *Gh* | GCTACAGACTCTCGGACCTC | GGAAAAGCACTAGCCTCCTGA |

*Appendix 1—table 2 Continued on next page*

Appendix 1—table 2 Continued

| | Primer sequences | |
|---|---|---|
| Target | Forward (5′ > 3′) | Reverse (5′ > 3′) |
| Ifng | GCCACGGCACAGTCATTGA | TGCTGATGGCCTGATTGTCTT |
| Igf1 | GTGGATGCTCTTCAGTTCGTGTG | TCCAGTCTCCTCAGATCACAGC |
| Igf1r | CGGGATCTCATCAGCTTCACAG | TCCTTGTTCGGAGGCAGGTCTA |
| Il11 | CTGACGGAGATCACAGTCTGGA | GGACATCAAGTCTACTCGAAGCC |
| Il1b | TGGACCTTCCAGGATGAGGACA | GTTCATCTCGGAGCCTGTAGTG |
| Il31 | ACAACGAAGCCTACCCTGGT | ACATCCCAGATGCCTGCTTTAT |
| Il6 | CACGGCCTTCCCTACTTCAC | TGCCATTGCACAACTCTTTTCT |
| Krt18 | ACTCCGCAAGGTGGTAGATGA | TCCACTTCCACAGTCAATCCA |
| Krt8 | TGGAAGGACTGACCGACGAGAT | GGCACGAACTTCAGCGATGAT |
| Lgr4 | CTCTCAACAACATCTCAAGCA | TAATTCAAGTCCAGGGTTTCCA |
| Lgr5 | GTAGGCAACCCTTCTCTTATCAC | CAAAGTCAGTGTTCTTAGTTCAGG |
| Lgr6 | CTCTCAACCATATCCGCCAC | TATAGTTCAGGTCTAGTGTCTCCA |
| Lif | CCCATGCCTGATACGCCTG | CAAGTGGGAGTGCAAACTGAC |
| Nog | GGTGGAGTTCAACATCCTGTGG | ATCCGCATCTCGTAGGCACTCA |
| Osm | ATGCAGACACGGCTTCTAAGA | TTGGAGCAGCCACGATTGG |
| Pou1f1 | ATGAGTTGCCAATCTTTCACC | TAATGAAGTCCTGTCGCTGTG |
| Pomc | AGAGGTTAAGAGCAGTGACTAAGAG | AACATGTTCAGTCTCCTGCCT |
| Prl | CTGGCTACACCTGAAGACAAG | CGAGGACTGCACCAAACTGA |
| Prop1 | CCATCTTTGGTTTGGGTGG | CAGAGCTCCTGTCTACCGT |
| Sox2 | AAAGTATCAGGAGTTGTCAAGG | CTCTTCTTTCTCCCAGCCC |
| Sox9 | CGGAACAGACTCACATCTCTCC | GCTTGCACGTCGGTTTTGG |
| Tacstd2 | GTCTGCCAATGTCGGGCAA | GTTGTCCAGTATCGCGTGCT |
| Tbx19 | AGCTGTGTCTACATTCACCC | TCAGCATTATCTGCCCACCT |
| Tnf | GGTGCCTATGTCTCAGCCTCTT | GCCATAGAACTGATGAGAGGGAG |
| Wnt5a | ATGCAGTACATTGGAGAAGGTG | CGTCTCTCGGCTGCCTATTT |

## Appendix 1—key resources table

| Reagent type (species) or resource | Designation | Source or reference | Identifiers | Additional information |
|---|---|---|---|---|
| Genetic reagent (*Mus musculus*) | *Sox2eGFP/+; Sox2tm1Lpev* | DOI: 10.1159/000082134. PMID:15711057. | MGI:3589809 | |
| Genetic reagent (*Mus musculus*) | *Il6−/−; Il6tm1Kopf* | DOI: 10.1038/368339a0. PMID:8127368. | MGI:1857197; RRID:IMSR_JAX:002650 | Gifted by Dr. Pura Muñoz Cánoves, University of Valencia |
| Genetic reagent (*Mus musculus*) | *GHCre/+; Tg(Gh1-cre)bKnmn* | DOI: 10.1210/en.2006–1542. PMID:17289844. | MGI:4442901 | |
| Genetic reagent (*Mus musculus*) | *ROSA26iDTR/+; Gt(ROSA)26Sortm1(HBEGF)Awai; −/ iDTR* | DOI: 10.1038/nmeth762. PMID:15908920. | MGI:3772576; RRID:IMSR_JAX:007900 | |
| Biological sample (*Mus musculus*) | Primary anterior pituitary (AP) cells | This paper | N/A | Freshly isolated from *Mus musculus* |
| Antibody | Anti-SOX2 (rabbit monoclonal) | Abcam | Cat. #:AB92494; RRID:AB_10585428 | (1:2000) |
| Antibody | Anti-SOX2 (goat polycolonal) | Immune Systems | Cat. #:GT15098; RRID:AB_2732043 | (1:750) |

*Appendix 1 Continued on next page*

*Appendix 1 Continued*

| Reagent type (species) or resource | Designation | Source or reference | Identifiers | Additional information |
|---|---|---|---|---|
| Antibody | Anti-ACTH (rabbit polyclonal) | National Hormone and Peptide Program (NHPP) | Cat. #:AFP-6328031; RRID:AB_2665562 | (1:5000) |
| Antibody | Anti-CK8/18 (guinea pig polyclonal) | Progen | Cat. #:GP11; RRID:AB_2904125 | (1:20) |
| Antibody | Anti-E-cad (rabbit monoclonal) | Cell Signalling Technology | Cat. #:3195; RRID:AB_2291471 | (1:400) |
| Antibody | Anti-GH (guinea pig polyclonal) | National Hormone and Peptide Program (NHPP) | Cat. #:AFP12121390;RRID: B_2756840 | (1:1000) |
| Antibody | Anti-GH (rabbit polyclonal) | National Hormone and Peptide Program (NHPP) | RRID:AB_2629219 | (1:10,000) |
| Antibody | Anti-Ki67 (rabbit monoclonal) | Thermo Fisher Scientific | Cat. #:RM-9106; RRID:AB_2341197 | (1:50) |
| Antibody | Anti-Ki67 (mouse monoclonal) | BD Bioscience | Cat. #:550609; RRID:AB_393778 | (1:100) |
| Antibody | Anti-PIT1 (rabbit polyclonal) | Other | Cat. #:422_Rhodes; RRID:AB_2722652 | (1:500) Gifted by Dr. S. J. Rhodes (IUPUI, USA); Dr. K. Rizzoti (Francis-Crick institute, UK) |
| Antibody | Anti-PRL (rabbit polyclonal) | National Hormone and Peptide Program (NHPP) | RRID:AB_2629220 | (1:10,000) |
| Antibody | Anti-TACSTD2 (goat polyclonal) | R&D systems | Cat. #:AF1122; RRID:AB_2205662 | (1:50) |
| Antibody | Anti-aGSU (guinea pig polyclonal) | National Hormone and Peptide Program (NHPP) | | (1:10,000) |
| Antibody | Anti-VIM (rabbit monoclonal) | Cell Signalling Technology | Cat. #:5741 S; RRID:AB_10695459 | (1:50) |
| Antibody | Anti-TdT (rabbit monoclonal) | Abcam | Cat. #:ab76544; RRID:AB_2094447 | (1:100) |
| Antibody | Anti-goat IgG 555 (donkey polyclonal) | Thermo Fisher Scientific | Cat. #:A-21432; RRID:AB_2535853 | (1:1000) |
| Antibody | Anti-mouse IgG 555 (donkey polyclonal) | Thermo Fisher Scientific | Cat. #:A-31570; RRID:AB_2536180 | (1:1000) |
| Antibody | Anti-mouse IgG 488 (donkey polyclonal) | Thermo Fisher Scientific | Cat. #:A-32766; RRID:AB_2762823 | (1:1000) |
| Antibody | Anti-rabbit IgG 488 (donkey polyclonal) | Thermo Fisher Scientific | Cat. #:A-21206; RRID:AB_2535792 | (1:1000) |
| Antibody | Anti-rabbit IgG 555 (donkey polyclonal) | Thermo Fisher Scientific | Cat. #:A-31572; RRID:AB_162543 | (1:1000) |
| Antibody | Anti-guinea pig FITC (donkey polyclonal) | Jackson ImmunoResearch | Cat. #:706-545-148; RRID:AB_2340472 | (1:500) |
| Sequence-based reagent | RNAscope probe *M. musculus Sox2* | Advanced Cell Diagnostics | Cat. #:401041 C3 | |
| Sequence-based reagent | RNAscope probe *M. musculus Rspo3* | Advanced Cell Diagnostics | Cat. #:483781 C2 | |
| Sequence-based reagent | RNAscope probe *M. musculus Lgr4* | Advanced Cell Diagnostics | Cat. #:318321 C2 | |
| Sequence-based reagent | RNAscope probe *M. musculus Lgr6* | Advanced Cell Diagnostics | Cat. #:404961 C2 | |
| Peptide, recombinant protein | R-Spondin 1; RSPO1 | R&D systems | Cat. #:4645-RS | |
| Peptide, recombinant protein | WNT3A | R&D systems | Cat. #:5036-WN | |
| Peptide, recombinant protein | Interleukin-6; IL-6 | Peprotech | Cat. #:200-06 | |
| Peptide, recombinant protein | Epidermal growth factor; EGF | R&D systems | Cat. #:236-EG | |
| Peptide, recombinant protein | Basic fibroblast growth factor; bFGF; FGF2 | R&D systems | Cat. #:234-FSE | |
| Peptide, recombinant protein | Insulin-like growth factor 1; IGF1 | Peprotech | Cat. #:100-11 | |
| Peptide, recombinant protein | Fibroblast growth factor 8; FGF8 | Peprotech | Cat. #:100-25 | |

*Appendix 1 Continued on next page*

*Appendix 1 Continued*

| Reagent type (species) or resource | Designation | Source or reference | Identifiers | Additional information |
|---|---|---|---|---|
| Peptide, recombinant protein | Fibroblast growth factor 10; FGF10 | Peprotech | Cat. #:100-26 | |
| Peptide, recombinant protein | Sonic hedgehog; SHH | Peprotech | Cat. #:464-SH | |
| Peptide, recombinant protein | Noggin | Peprotech | Cat. #:120-10C | |
| Peptide, recombinant protein | IL-1β | Peprotech | Cat. #:200-01B | |
| Peptide, recombinant protein | Tumor necrosis factor-α; TNFα | Peprotech | Cat. #:300-01A | |
| Peptide, recombinant protein | Leukemia-inhibitory factor; LIF | Peprotech | Cat. #:300-05 | |
| Peptide, recombinant protein | IL-11 | Peprotech | Cat. #:200-11 | |
| Peptide, recombinant protein | WNT5A | AMSBIO | Cat. #:AMS.P5172 | |
| Peptide, recombinant protein | RSPO3 | Peprotech | Cat. #:120-44 | |
| Commercial assay or kit | RNAscope Multiplex Fluorescent Reagent Kit v2 | Advanced Cell Diagnostics | Cat. #:323,100 | |
| Commercial assay or kit | RNeasy Micro kit | Qiagen | Cat. #:74,004 | |
| Commercial assay or kit | V-PLEX mouse IL-6 kit | Meso Scale Discovery | Cat. #:K152QXD-1 | |
| Commercial assay or kit | Superscript III First-Strand Synthesis Supermix | Invitrogen | Cat. #:18080400 | |
| Commercial assay or kit | SYBR Green PCR Master Mix | Thermo Fisher Scientific | Cat. #:4309155 | |
| Chemical compound, drug | LGK-974; LGK | Biogems | Cat. #:1241454 | |
| Chemical compound, drug | Diphtheria toxin; DT | Sigma-Aldrich | Cat. #:D0564 | |
| Chemical compound, drug | Cholera toxin; CT | Sigma-Aldrich | Cat. #:C8052 | |
| Chemical compound, drug | SB202190; P38i | Sigma-Aldrich | Cat. #:S7067 | |
| Chemical compound, drug | A83-01 | Sigma-Aldrich | Cat. #:SML0788 | |
| Chemical compound, drug | Nicotinamide | Sigma-Aldrich | Cat. #:72,340 | |
| Chemical compound, drug | N-Acetyl-cysteine | Sigma-Aldrich | Cat. #:A7250 | |
| Chemical compound, drug | B-27 | Thermo Fisher Scientific | Cat. #:12587010 | |
| Chemical compound, drug | L-Glutamine | Thermo Fisher Scientific | Cat. #:25030081 | |
| Chemical compound, drug | N2 | Thermo Fisher Scientific | Cat. #:17502048 | |
| Chemical compound, drug | STATTIC | Merck | Cat. #:573,099 | |
| Chemical compound, drug | XAV939; XAV | Merck | Cat. #:575,545 | |
| Chemical compound, drug | IWP2 | Merck | Cat. #:681,671 | |

*Appendix 1 Continued on next page*

*Appendix 1 Continued*

| Reagent type (species) or resource | Designation | Source or reference | Identifiers | Additional information |
|---|---|---|---|---|
| Chemical compound, drug | Matrigel | Corning | Cat. #:356,234 | |
| Software, algorithm | GraphPad Prism Software | GraphPad Prim (https://www.graphpad.com/) | RRID:SCR_002798 | Version 9.3.1 |
| Software, algorithm | Fiji | ImageJ (http://imagej.nih.gov/ij/) DOI: 10.1038/nmeth.2019 PMID:22743772 | RRID:SCR_002285 | |
| Software, algorithm | IncuCyte software | Sartorius (https://www.sartorius.com/en/products/live-cell-imaging-analysis/live-cell-analysis-software/incucyte-s3-software-v2018b) | | Version 2018B |
| Software, algorithm | MSD discovery workbench software | MSD (https://www.mesoscale.com/en/products_and_services/software) | RRID:SCR_019192 | Version 4.0.12 |
| Other | Deposited Data, scRNA-seq | ArrayExpress | EMBL-EBI:E-MTAB-11337 | scRNA-seq of neonatal mouse AP in steady-state conditions and after transgenically inflicted local damage (see Materials and methods) |

