## [Editor Report]

This is a well conducted study on development of neonatal mouse pituitary using multiple ScRNA Seq and organoid culture models. In this revised version, the authors have done a thorough job of addressing the concerns of the reviewers on the previous version and significantly improved it. In particular they have provided additional data with regard to the role of Wnt signaling and further defined the IL6 null mouse phenotypes.

---

## [Decision Letter]

**Decision letter after peer review:**

Thank you for submitting your article "Decoding the activated stem cell phenotype of the vividly maturing neonatal pituitary" for consideration by *eLife*. Your article has been reviewed by 2 peer reviewers, and the evaluation has been overseen by a Reviewing Editor and Nancy Carrasco as the Senior Editor. The following individual involved in review of your submission has agreed to reveal their identity: Gwen Childs (Reviewer #2).

Essential revisions:

1) Want pathway involvement with regard to data presented in Figure 2 must be tested.

2) Demonstration of other pituitary defects in IL6 null mice need to be reported.

3) New folliculo-stellate markers have been characterised in the mouse and should be tested to clarify the identity of MC cluster. what are the MC cells in both the murine and human datasets?

4) Data on the mechanisms of regeneration following toxin ablation of GH cells needs to be defined.

5) The language of terms used in the Text as pointed out by Rev 2 must be addressed.

*Reviewer #1 (Recommendations for the authors):*

The results presented in this manuscript are interesting, however further investigation would greatly improve their significance. Furthermore, I have some precise questions about some of the data.

What is the sex of the animal used in this study? If both sexes were used, was any potential sex difference examined? If only one sex was used, were the main conclusions tested on the other sex?

Co-cultures of young organoids with older ones increase proliferative abilities of the latter (Figure 2). The authors suggest that Wnt signalling may be involved but this could be tested using conditioned medium, and potentially manipulating Wnt pathway.

Removal of growth factors in the organoid medium does not affect growth of adult organoids but improve number of neonatal ones. Are these factors superfluous for adult organoids? The data clearly shows that they actively inhibit neonatal cells. I think the significance of these observations should be further explored and discussed. Is there any evidence that these pathways are active, or regulators expressed in the young but not the old stem cells?

Supplementation of Il6 null organoids with LIF or IL-11 appears to rescue organoid formation. Do they also improve self-renewal (passage number)? Is there a dual role of the JAK-STAT pathway on organoid formation efficiency and self-renewal/passageability? I found the interpretation of these data (line 233-245) difficult to follow. A clear distinction between organoid formation efficiency and passage number could be made to clarify the conclusions of the authors.

I understand that pituitary stem cell proliferation is normal in IL6 null glands. Did the authors examine for the presence of further pituitary defects?

The authors identify the MC cluster as folliculo-stellate cells. These were previously shown to express *SOX2*. Here this is clearly not the case (Figure 1B). In addition, in their human single-cell study (Zhang et al., 2020), the authors state that folliculo-stellate cells are not present. Therefore, the identity of the MC cluster in the re-analysis of their dataset (Figure 3G), that I could not find in the original study, is unclear to me. I would have expected the cluster identification to be conserved, so more detail is needed here such as the method used for the identification of the different clusters. New folliculo-stellate markers have been characterised in the mouse and should be tested here to clarify this issue: what are the MC cells in both the murine and human datasets? This deserves further investigation because of the potential Wnt-mediated crosstalk with stem cells suggested, but not substantiated, in the present study.

Following DT mediated GH-cell ablation, regeneration is observed. Because this system is Cre-mediated, it is difficult to perform lineage tracing and identify the cells generating the new somatotrophs. However, mechanisms leading to this regeneration are not further investigated. A potential role for Wnt signalling, in the context of earlier results, could have been examined?

Davis et al., 2016 should be replaced by Perez-Millan et al., (line 153).

*Reviewer #2 (Recommendations for the authors):*

1. My concerns focus mainly on the grammar and context. The science is outstanding. The availability of data is outstanding.

2. Line 40 and Title. I am not certain the word vividly is used in the right context. The English meaning of vividly is brightly or intensely, clear, distinct, powerful, detailed, strongly, perceptibly, sharply, graphically. None of those synonyms would be appropriate as a way to describe neonatal development. Graphically is the only synonym that might be appropriate and indeed might be more appropriate than vividly. I am looking for another meaning that might fit with a bioinformatics type analysis and don't find one. There is no question that this is neonatal development, so why add an adverb that means "brightly, intensely, powerful, strongly, distinctly, sharply, perceptibly….etc. If you mean graphically, then that should be used, but even that doesn't describe "maturation".

3. Lines 83-100 and Figure 1. It appears that the posterior lobe was removed before cell dispersion. If so, that would have removed the intermediate lobe. However, melanotropes are still present from both groups, so it is not surprising that the PL cells are still there as well. Was the PL removed?

4. Line 131. The use of the word vivid is correct in this line as it points to powerful, distinct, detailed, etc. based on the findings.

5. Line 143. It is unclear what is meant by "In line"

6. Figure 1, bottom. Which is the neonatal pituitary and which is the adult pituitary in these immunolabeled fields.

7. Line 327. The English use of the word vivid is not appropriate in this context. What is meant by "vivid neonatal" pituitary. Of course one can see it clearly and it is developing, however vivid or even graphic is inappropriate here. It is simply a developing pituitary. Vivid only refers to the fact that you can perceive or see it intensely. Just discuss it as "developing" neonatal pituitary. Your analysis was robust and intense and comprehensive, however that describes the analysis, not the "maturation".

8. Line 372. The concept "vividly maturing" is not comprehensible in English. One would never use a synonym of vividly, like "brightly maturing, or distinctly maturing, or intensely maturing". The antonym for vivid would mean "unable to be seen or indistinct". Clearly you can see maturation with a number of approaches, however just describing it by the fact that it can be seen "brightly" or distinctly" is not descriptive. Unless there is a bioinformatics use of the word vividly of which I am unaware, another word should be chosen wherever it is used. One could use the word "actively maturing", however because only one age was chosen, the dynamics of maturation must be assumed based on PN7. I suggest that the term vividly be omitted as it is not a descriptive adverb.

---

## [Author Response]

Essential revisions:1) Want pathway involvement with regard to data presented in Figure 2 must be tested.

To address this interesting request, and investigate whether WNT ligands, secreted by neonatal pituitary organoids mediate, at least partly, the increased proliferative growth of adult pituitary organoids when cultured together in a Matrigel droplet, we tested the following approaches.

First, as suggested by Reviewer 1, we collected conditioned medium (CM) from neonatal pituitary organoids and assessed whether expression of WNT target genes is increased in adult (fully-grown) pituitary organoids when exposed to this CM (Author response image 1). The outcome of this approach was not conclusive: although *Lgr5* expression appeared to be upregulated, the other universal WNT target gene *Axin2* was not (Author response image 1). Other target genes (such as *Lgr6*) were more variable. Although the Reviewer’s suggestion looked straightforward, a literature search did actually not reveal studies with such CM approach to show WNT ligand secretion and activity. Indeed, it is known that WNT ligands are secreted by cells in only minute quantities (Mikels et al., 2006); and are very short-range signals, not traveling far in tissue (Routledge et al., 2019), moreover being hydrophobic and getting associated with the ECM (Mikels et al., 2006). Hence, secreted WNT ligands, while able to affect neighboring cells in the close environment (as in the co-culture situation of neonatal + adult organoids), are expected not to be capable of leaving the Matrigel droplet at reasonable levels, moreover being highly diluted in the medium if a small minority of ligands would eventually be able to enter the CM.

**Author response image 1. sa2fig1:** (**A**) Schematic overview of the organoid CM experiment. (**B**) Representative brightfield images of adult AP organoids exposed to CM or not (CTRL). Scale bar, 500 µm. (**C**) Gene expression analysis for main WNT target genes in adult AP organoids exposed to CM, compared to CTRL, set as 1 (dashed line). Bars represent mean ± SD (n=2).

Therefore, we tested another approach to answer the reviewer’s question, i.e. using the physiological WNT signaling inhibitor Dickkopf 1 (DKK1) (which binds to the WNT co-receptors LRP5/6 thereby blocking WNT ligand action). Adding this inhibitor to neonatal (WT) + adult (tdT^+^) AP organoid co-cultures resulted in decreased proliferative activity (% Ki67^+^ cells) in the adult organoids These new findings strongly support that the rise in proliferation in adult pituitary organoids, when cultured together with neonatal pituitary organoids, is mediated, at least partly, by WNT (paracrine) activity. We now included these new data in the Results section (lines 317-322), and added them to Figure 3 —figure supplement 2F.

2) Demonstration of other pituitary defects in IL6 null mice need to be reported.

In answer to this understandable question of Reviewer 1, we would like to clarify that the full examination and phenotyping of IL6 null pituitary (at multiple ages in both homeostatic and perturbed conditions) will be part of a comprehensive, forthcoming publication that we are preparing. In short, in neonatal IL6 null pituitary, no major phenotype regarding stem, progenitor and endocrine cells is observed in homeostatic conditions. To already touch upon this in the current study (without including and revealing all detailed analyses meant to be incorporated in the new paper), we provide extra gene expression data of stem, progenitor and endocrine cell markers in the neonatal IL6 KO *versus* WT pituitary, showing no differences (see lines 235-237 and new Figure 2 —figure supplement 3B).

3) New folliculo-stellate markers have been characterised in the mouse and should be tested to clarify the identity of MC cluster. what are the MC cells in both the murine and human datasets?

The historically designated folliculo-stellate (FS) cells have eventually been found to represent a heterogeneous cell population in the pituitary, encompassing, among others, paracrine-regulatory, mesenchymal supportive as well as stem cells. Indeed, mapping the newly proposed FS cell markers (as reported by (Ho et al., 2020) and by (Fletcher et al., 2021) (rat pituitary) on our dataset revealed enriched expression in the mesenchymal cell (MC) and stem cell clusters (Author response image 2 and Author response image 3) and in pericytes (markers from (Ho et al., 2020); Author response image 2), cells being a general part of the mesenchymal component of tissues. Thus, it is clear (thereby supporting many previous observations) that FS cells do not form a delineated separate cell cluster but that this ‘flag covers multiple cargos’, being distributed over the stem cell and MC clusters. Comparable findings are observed in the human fetal pituitary dataset (Zhang et al., 2020) (Author response image 2 and Author response image 3). Other scRNA-seq papers on mouse pituitary did also not identify (designate) a separate FS cell cluster (Cheung et al., 2018; Ruf-Zamojski et al., 2021; Vennekens et al., 2021).

**Author response image 2. sa2fig2:** FS cell markers from (Ho et al., 2020). These markers are mainly found in the MC population of mouse and human pituitary (and in the pericytes). *Col3a1*, *Itih5* and *Col1a1* were also identified and used in (Cheung et al., 2018) and (Vennekens et al., 2021) to annotate the CT cluster (thus, the MC cluster in our paper, see Figure 1 —figure supplement 1B).

**Author response image 3. sa2fig3:** FS cell markers from (Fletcher et al., 2022) (rat pituitary). These markers are mainly found in the neonatal AP stem cell and progenitor cell populations. As these markers are also present in many other cell clusters (as also apparent in the human fetal pituitary data), they do not seem to be highly specific for FS cells.

Regarding the MC cluster, its annotation is based on the mesenchymal markers published in (Cheung et al., 2018) and in (Vennekens et al., 2021) (in which this cluster was named ‘connective tissue’ (CT) cluster) (see Supplementary Figure 1 —figure supplement 1B). These cells are considered to represent the supportive MC of the gland (both physically and regulatory).

Additional newly defined markers of FS cells include *S100a6* (mouse; (Vennekens et al., 2021); shown in Figure 1 —figure supplement 1B), which is expressed in both stem cell clusters and MC of the neonatal AP; and Aldolase-C (*Aldoc;* rat and mouse; (Fujiwara et al., 2020)) which shows high expression in the stem cell clusters of the mouse neonatal AP (but is expressed throughout the human fetal pituitary cell types) (Author response image 4).

**Author response image 4. sa2fig4:** Aldolase C (*Aldoc*) expression, mapped on our neonatal mouse AP atlas (left) and the human fetal pituitary atlas (right).

Taken together, we feel that the cluster identification of our dataset has been done in an appropriate manner, concordant with several other pituitary scRNA-seq studies.

4) Data on the mechanisms of regeneration following toxin ablation of GH cells needs to be defined.

In our originally submitted manuscript, we already included two potential mechanisms: (i) proliferation of GH^+^ and PIT1^+^ progenitor cells (Figure 4F); and (ii) differentiation of *sox2*^+^ stem cells towards new GH^+^ somatotropes, i.e. increase in GH/*SOX2* co-expressing cells (for which we now performed extra experiments to n=4 (new Figure 4G)); and (iii) decreased NOTCH signaling in damaged pituitary stem cells (Figure 4 —figure supplement 2B).

Moreover, we also already mentioned that both WNT and IL-6 signaling are most likely not involved in regeneration of the neonatal AP. *Il6* expression is not increased in the stem cells or MC following the local damage (Figure 4 —figure supplement 1D), which is in clear contrast to findings in the adult pituitary (Vennekens et al., PNAS (2021)). Similarly, expression of WNT pathway components, neither the target genes *Lgr5* and *Axin2* (now added to the heatmap in Figure 4 —figure supplement 1E), did not acutely change after the damage infliction in the AP (Figure 4 —figure supplement 1E). To further examine a possible role of WNT in regeneration, we performed additional RT-qPCR analyses on damaged (regenerating) and control pituitary for the WNT target genes at later timepoints, i.e. one (PD14) and two weeks (PD21) after damage infliction. No upregulation was observed. These new data were included in the Results section (lines 370-373) and figures (Figure 4 —figure supplement 1E).

5) The language of terms used in the Text as pointed out by Rev 2 must be addressed.

We replaced the word ‘vivid/vividly’ with ‘active/dynamic’ (or other appropriate words) at the indicated places, or just deleted it (e.g. in the title and Abstract).

Reviewer #1 (Recommendations for the authors):The results presented in this manuscript are interesting, however further investigation would greatly improve their significance. Furthermore, I have some precise questions about some of the data.What is the sex of the animal used in this study? If both sexes were used, was any potential sex difference examined? If only one sex was used, were the main conclusions tested on the other sex?

For the neonatal pituitary experiments, pups were taken (and pooled) irrespective of sex since it is hard to correctly score sex at this very early age (PD7). In preparatory analyses, we did not find substantial differences between male and female PD7 pituitary regarding stem/progenitor and endocrine cells. In particular, the gonadotrope (sex hormone) lineage does not look different between both sexes (number of LHb^+^ and FSHb^+^ cells (Author response image 5) and *Lhb* and *Fshb* gene expression level (Author response image 5)). Moreover, proportion of *sox2*^+^ cells and expression of stem/progenitor cell markers is similar (Author response image 5).

**Author response image 5. sa2fig5:** A. Proportion of indicated cell types in the AP of male and female PD7 mice. B. Expression of indicatedgenes in the AP of male and female PD7 mice. (NB. sex determination by PCR amplification of the Y chromosome specific gene Kdm5d). C. DEG associated GO terms linked with NOTCH signaling enriched in SC1, SC2 and Prolif SC, in separate comparison of damaged male and female AP versus control (corresponding to Figure 4 - figure supplement 2B, in which both sexes are combined for comparison).

We retrospectively observed that, in the scRNA-seq analyses, one damaged pituitary was from male and one from female origin (based on *Xist* expression in the scRNA-seq analysis). We repeated some of the previous comparisons with control pituitary (such as DEG-associated GO analysis between the stem cell clusters, as presented in Figure 4 —figure supplement 2) for the individual sexes separately and did not find differences compared to our original conclusions (see e.g. for NOTCH in Author response images 5 and 5).

For the comparison experiments with adult pituitary (such as organoid analyses), biological replicates were performed with male or female adult mice, and results (which were found comparable irrespective of sex) combined for statistical analysis. We now added a sentence clarifying the sex of the animals used (lines 465-468).

Co-cultures of young organoids with older ones increase proliferative abilities of the latter (Figure 2). The authors suggest that Wnt signalling may be involved but this could be tested using conditioned medium, and potentially manipulating Wnt pathway.

Please see answer above (point 1 of Essential revisions).

Removal of growth factors in the organoid medium does not affect growth of adult organoids but improve number of neonatal ones. Are these factors superfluous for adult organoids? The data clearly shows that they actively inhibit neonatal cells. I think the significance of these observations should be further explored and discussed. Is there any evidence that these pathways are active, or regulators expressed in the young but not the old stem cells?

We thank the reviewer for this interesting suggestion to look closer into these pathways in neonatal and adult stem cells.

For the reviewer’s information, removal of (many of) these growth factors does negatively affect growth of adult organoids as shown in our previous paper in which the organoid model was established (Cox et al., 2019). In our current study, we found that several of these growth factors are not needed (rather impedimental) for neonatal pituitary organoid growth. To address the suggestion of the reviewer, we analyzed whether corresponding factors/regulators are indeed higher expressed in neonatal than adult stem cells (which thus may explain why they are not needed for neonatal organoid growth). Indeed, among others, *Fgf10* and *Igf1,* together with their receptors *Fgfr2* and *Igf1r*, as well as noggin (*Nog*), are found higher expressed in the neonatal than adult stem cells, as extracted from the scRNA-seq data. Moreover, these expression differences remain present when the stem cells are grown as organoids. These new data are now mentioned in the text (lines 176-181) and included as new figure (Figure 2 —figure supplement 1B).

Supplementation of Il6 null organoids with LIF or IL-11 appears to rescue organoid formation. Do they also improve self-renewal (passage number)?

To address this interesting question, we performed new experiments and found that addition of LIF and IL-11 indeed prolongs the expandability of the IL-6^-/-^ AP organoids. At the end of this revision period, they were still in culture (thus, passageable beyond P4) while not in the other conditions tested (including RSpECT), in which the IL-6^-/-^ AP organoids are lost in earlier passages. We included these new findings in the text (lines 246-250) and in Figure 2 —figure supplement 3E.

Is there a dual role of the JAK-STAT pathway on organoid formation efficiency and self-renewal/passageability? I found the interpretation of these data (line 233-245) difficult to follow. A clear distinction between organoid formation efficiency and passage number could be made to clarify the conclusions of the authors.

To answer this question, we performed new experiments in which the STAT3 inhibitor STATTIC was added at passaging (P0-P1), and found that organoids did not regrow, thereby supporting a role for JAK/STAT signaling in both organoid initial development (as already described in the original manuscript) and passaging (as added now to Figure 2 —figure supplement 3G). We also specified this better in the text (lines 250-252).

I understand that pituitary stem cell proliferation is normal in IL6 null glands. Did the authors examine for the presence of further pituitary defects?

Please see answer above (point 2 of Essential revisions).

The authors identify the MC cluster as folliculo-stellate cells. These were previously shown to express SOX2. Here this is clearly not the case (Figure 1B). In addition, in their human single-cell study (Zhang et al., 2020), the authors state that folliculo-stellate cells are not present. Therefore, the identity of the MC cluster in the re-analysis of their dataset (Figure 3G), that I could not find in the original study, is unclear to me. I would have expected the cluster identification to be conserved, so more detail is needed here such as the method used for the identification of the different clusters.

For the definition/description of folliculo-stellate (FS) cells and their relation to MC (and other cell types including stem cells), please see already our answer above (point 3 of Essential revisions). In particular, FS cells represent a heterogeneous cell population encompassing, among others, supportive cells (MC) and stem cells. In other words, the MC population is a part of the heterogeneous FS cell population, as also the stem cell population is. Hence, *SOX2*, indeed previously reported to be expressed by the (heterogeneous) FS cell population, is expressed in its stem cell part, whereas the MC part expresses mesenchymal (but not stem cell) markers. In our manuscript, we do not ‘identify the MC cluster as FS cells’, but mention that the MC cluster is a part of the formerly designated FS population (lines 280-283). Of note, the term FS cells is used in literature in ambiguous ways: (Ho et al., 2020) designate an FS cell cluster not containing stem cells (thus, no co-expression of *Sox2* and ‘FS cell’ markers), whereas (Fletcher et al., 2021) report clear co-expression, all together pointing to a certain level of ambiguity in the definition and marker profile of FS cells (or, ‘what’s in a name?’). By the way, also (Zhang et al., 2020) report the FS cell marker *S100B* in some of the stem cells, not different from the S100B-negative stem cells regarding DEGs.

Regarding the MC cluster in the human scRNA-seq dataset of (Zhang et al., 2020), in supplementary figure 1D of the publication which shows the entire UMAP of all >4,000 sequenced cells, the MC cluster is indicated. However, in the main text, the authors state that ‘nine clusters of anterior pituitary endocrine cells including the stem cells (Stem), cycling cells (CC), corticotropes, progenitors of the PIT-1 lineage (Pro.PIT1), somatotropes, lactotropes, thyrotropes, precursors of gonadotropes (Pre.Gonado) and gonadotropes, comprising 2,388 cells*’* were used for further analysis, causing the fact that the MC cluster is not visible in the main figures (as the reviewer noticed). However, here in our present analyses, we used and analyzed the entire dataset of (Zhang et al., 2020), therefore also revealing the MC cluster.

New folliculo-stellate markers have been characterised in the mouse and should be tested here to clarify this issue: what are the MC cells in both the murine and human datasets? This deserves further investigation because of the potential Wnt-mediated crosstalk with stem cells suggested, but not substantiated, in the present study.

Please see answer above (point 3 of Essential revisions). The potential crosstalk, as inferred from CellPhoneDB analysis (Figure 3D), will in follow-up studies indeed be comprehensively examined, thereby needing extensive genetic mouse and (composite) organoid models.

Following DT mediated GH-cell ablation, regeneration is observed. Because this system is Cre-mediated, it is difficult to perform lineage tracing and identify the cells generating the new somatotrophs. However, mechanisms leading to this regeneration are not further investigated. A potential role for Wnt signalling, in the context of earlier results, could have been examined?

Please see answer above (point 4 of Essential revisions).

Davis et al., 2016 should be replaced by Perez-Millan et al., (line 153).

We thank the reviewer for this attentive remark, and replaced the reference Davis et al., 2016 with ‘*PROP1* triggers epithelial-mesenchymal transition-like process in pituitary stem cells. Perez-Milan, et al., *eLife* (2016).’

Reviewer #2 (Recommendations for the authors):1. My concerns focus mainly on the grammar and context. The science is outstanding. The availability of data is outstanding.

We thank the reviewer for the positive appraisal of our work.

2. Line 40 and Title. I am not certain the word vividly is used in the right context. The English meaning of vividly is brightly or intensely, clear, distinct, powerful, detailed, strongly, perceptibly, sharply, graphically. None of those synonyms would be appropriate as a way to describe neonatal development. Graphically is the only synonym that might be appropriate and indeed might be more appropriate than vividly. I am looking for another meaning that might fit with a bioinformatics type analysis and don't find one. There is no question that this is neonatal development, so why add an adverb that means "brightly, intensely, powerful, strongly, distinctly, sharply, perceptibly….etc. If you mean graphically, then that should be used, but even that doesn't describe "maturation".

We used the word ‘vivid/vividly’ to indicate the actively (‘powerfully’, ‘intensely’) maturing state of the pituitary at this neonatal age. However, we understand the reviewer’s concern and that ‘vivid’ may not be the most appropriate word. Therefore, we removed it (e.g. in title and Abstract) or replaced it throughout the manuscript with ‘active/dynamic’ (or other appropriate words) at the indicated places.

3. Lines 83-100 and Figure 1. It appears that the posterior lobe was removed before cell dispersion. If so, that would have removed the intermediate lobe. However, melanotropes are still present from both groups, so it is not surprising that the PL cells are still there as well. Was the PL removed?

Please see answer above (point 4).

4. Line 131. The use of the word vivid is correct in this line as it points to powerful, distinct, detailed, etc. based on the findings.

In analogy with the replacement of ‘vivid’ at other places, we also changed the word here.

5. Line 143. It is unclear what is meant by "In line"

‘In line’ was used as ‘in accordance’. We changed this now (line 110, 146 and 378).

6. Figure 1, bottom. Which is the neonatal pituitary and which is the adult pituitary in these immunolabeled fields.

The designation of ‘neonatal’ and ‘adult’ was shown above the heatmaps of this figure. However, since not clear, we repeat the terms now above the immunofluorescence images.

7. Line 327. The English use of the word vivid is not appropriate in this context. What is meant by "vivid neonatal" pituitary. Of course one can see it clearly and it is developing, however vivid or even graphic is inappropriate here. It is simply a developing pituitary. Vivid only refers to the fact that you can perceive or see it intensely. Just discuss it as "developing" neonatal pituitary. Your analysis was robust and intense and comprehensive, however that describes the analysis, not the "maturation".

We understand the reviewer’s concern and agree that ‘vivid’ may not be the most appropriate word (although it sounded so when translated to our native language). Therefore, we removed it (e.g. in title and Abstract) or replaced it throughout the manuscript with ‘active/dynamic’ (or other appropriate words) at the indicated places.

8. Line 372. The concept "vividly maturing" is not comprehensible in English. One would never use a synonym of vividly, like "brightly maturing, or distinctly maturing, or intensely maturing". The antonym for vivid would mean "unable to be seen or indistinct". Clearly you can see maturation with a number of approaches, however just describing it by the fact that it can be seen "brightly" or distinctly" is not descriptive. Unless there is a bioinformatics use of the word vividly of which I am unaware, another word should be chosen wherever it is used. One could use the word "actively maturing", however because only one age was chosen, the dynamics of maturation must be assumed based on PN7. I suggest that the term vividly be omitted as it is not a descriptive adverb.

When using the word ‘vivid’, we meant ‘dynamic’ or ‘active’. We now changed it accordingly.

References

Cheung, L. Y. M., George, A. S., McGee, S. R., Daly, A. Z., Brinkmeier, M. L., Ellsworth, B. S., and Camper, S. A. (2018). Single-cell RNA sequencing reveals novel markers of male pituitary stem cells and hormone-producing cell types. *Endocrinology*, *159*(12), 3910–3924. doi: 10.1210/en.2018-00750

Cox, B., Laporte, E., Vennekens, A., Kobayashi, H., Nys, C., Van Zundert, I., Uji-i, H., Vercauteren Drubbel, A., Beck, B., Roose, H., Boretto, M., and Vankelecom, H. (2019). Organoids from pituitary as novel research model to study pituitary stem cell biology. *Journal of Endocrinology*, *240*(2), 287–308. doi: 10.1530/JOE-18-0462

Fletcher, P. A., Prévide, R. M., Smiljanic, K., Sherman, A., Coon, S. L., and Stojilkovic, S. S. (2021). Transcriptomic heterogeneity of Sox2-expressing pituitary cells. *BioRxiv*, 2021.12.10.472137. doi: 10.1101/2021.12.10.472137

Fujiwara, K., Tsukada, T., Horiguchi, K., Fujiwara, Y., Takemoto, K., Nio-Kobayashi, J., Ohno, N., and Inoue, K. (2020). Aldolase C is a novel molecular marker for folliculo-stellate cells in rodent pituitary. *Cell and Tissue Research*, *381*(2), 273–284. doi: 10.1007/S00441-020-03200-1

Gremeaux, L., Fu, Q., Chen, J., and Vankelecom, H. (2012). Activated phenotype of the pituitary stem/progenitor cell compartment during the early-postnatal maturation phase of the gland. *Stem Cells and Development*, *21*(5), 801–813. doi: 10.1089/scd.2011.0496

Hemeryck, L., Hermans, F., Chappell, J., Kobayashi, H., Lambrechts, D., Lambrichts, I., Bronckaers, A., and Vankelecom, H. (2022). Organoids from human tooth showing epithelial stemness phenotype and differentiation potential. *Cellular and Molecular Life Sciences : CMLS*, *79*(3), 153. doi: 10.1007/S00018-022-04183-8

Ho, Y., Hu, P., Peel, M. T., Chen, S., Camara, P. G., Epstein, D. J., Wu, H., and Liebhaber, S. A. (2020). Single-cell transcriptomic analysis of adult mouse pituitary reveals sexual dimorphism and physiologic demand-induced cellular plasticity. *Protein and Cell*, 1–19. doi: 10.1007/s13238-020-00705-x

Mikels, A. J., and Nusse, R. (2006). Wnts as ligands: processing, secretion and reception. *Oncogene 2006 25:57*, *25*(57), 7461–7468. doi: 10.1038/sj.onc.1210053

Routledge, D., and Scholpp, S. (2019). Mechanisms of intercellular wnt transport. *Development (Cambridge)*, *146*(10). doi: 10.1242/dev.176073

Ruf-Zamojski, F., Zhang, Z., Zamojski, M., Smith, G. R., Mendelev, N., Liu, H., Nudelman, G., Moriwaki, M., Pincas, H., Castanon, R. G., Nair, V. D., Seenarine, N., Amper, M. A. S., Zhou, X., Ongaro, L., Toufaily, C., Schang, G., Nery, J. R., Bartlett, A., … Sealfon, S. C. (2021). Single nucleus multi-omics regulatory landscape of the murine pituitary. *Nature Communications*, *12*(1), 2677. doi: 10.1038/s41467-021-22859-w

Russell, J. P., Lim, X., Santambrogio, A., Yianni, V., Kemkem, Y., Wang, B., Fish, M., Haston, S., Grabek, A., Hallang, S., Lodge, E. J., Patist, A. L., Schedl, A., Mollard, P., Nusse, R., and Andoniadou, C. L. (2021). Pituitary stem cells produce paracrine WNT signals to control the expansion of their descendant progenitor cells. *eLife*, *10*(1), e59142. doi: 10.7554/eLife.59142

Vennekens, A., Laporte, E., Hermans, F., Cox, B., Modave, E., Janiszewski, A., Nys, C., Kobayashi, H., Malengier-Devlies, B., Chappell, J., Matthys, P., Garcia, M. I., Pasque, V., Lambrechts, D., and Vankelecom, H. (2021). Interleukin-6 is an activator of pituitary stem cells upon local damage, a competence quenched in the aging gland. Proceedings of the National Academy of Sciences of the United States of America, 118(25), e2100052118. doi: 10.1073/pnas.2100052118

Zhang, S., Cui, Y., Ma, X., Yong, J., Yan, L., Yang, M., Ren, J., Tang, F., Wen, L., and Qiao, J. (2020). Single-cell transcriptomics identifies divergent developmental lineage trajectories during human pituitary development. *Nature Communications*, *11*(1), 5275. doi: 10.1038/s41467-020-19012-4